

**Temporal changes in photoreactivity of dissolved organic carbon and implications for aquatic**
**carbon fluxes from peatlands**
Amy E Pickard[1, 2*], Kate V Heal[1], Andrew R McLeod[1] and Kerry J Dinsmore[2]
[1]School of GeoSciences, Alexander Crum Brown Rd, Kings Buildings, The University of Edinburgh,
UK, EH9 3FF *amy.pickard@ed.ac.uk
[2]Centre for Ecology & Hydrology, Bush Estate, Penicuik, UK, EH26 0QB
**Abstract**
Aquatic systems draining peatland catchments receive a high loading of dissolved organic carbon
(DOC) from the surrounding terrestrial environment. Whilst photo-processing is known to be an
important process in the transformation of aquatic DOC, the drivers of temporal variability in this
pathway are less well understood. In this study, laboratory irradiation experiments were conducted on
water samples collected from two contrasting peatland aquatic systems in Scotland. The first system
was a stream draining the Auchencorth Moss peatland with high DOC concentrations subject to
strong seasonal and flow driven variability. The second was the low DOC reservoir, Loch Katrine,
also situated in a catchment with a high percentage peat cover. Samples were collected monthly at
both sites from May 2014 to May 2015 and from the stream system during two rainfall events. DOC
concentrations, absorbance properties and fluorescence characteristics were measured to investigate
characteristics of the photochemically labile fraction of DOC. $CO_2$ and CO produced by irradiation
were also measured to determine total photoproduction and intrinsic sample photoreactivity.
Significant variation was seen in the photoreactivity of DOC between the two systems, with total
irradiation induced changes typically two orders of magnitude greater at the high DOC stream site.
This is attributed to longer water residence times in the reservoir rendering a higher proportion of the
DOC recalcitrant to photo-processing. Rainfall events were identified as important in replenishing
photoreactive material in the stream, with lignin phenol data ($Ad:Al_{v,s}$ and P:V) indicating
mobilisation of fresh DOC derived from woody vegetation in the upper catchment. Using DOC-$CO_2$
conversion data from irradiation experiments, we estimate that the contribution of Auchencorth Moss



catchment to photo-induced aquatic $CO_2$ production is up to $3.48 \pm 2.02$ kg $CO_2$ yr$^{-1}$. We have shown
that peatland catchments produce significant volumes of aromatic DOC and that photoreactivity of
this DOC is greatest in the headwaters, however an improved understanding of water residence times
and DOC input-output along the source to sea aquatic pathway is required to determine the fate of
peatland carbon.
**Keywords:** Carbon budgets ▪ Rainfall events ▪ Lignin phenols
**1.  Introduction**
DOC is transported from terrestrial environments to aquatic systems where it plays an important role
in carbon (C) cycling. Biogeochemical transformations of DOC via microbial and photochemical
pathways impact significantly on aquatic C cycles, with up to 55% of C exported as DOC to
freshwaters estimated to be lost to the atmosphere as $CO_2$ (Cole et al., 2007; Tranvik et al., 2009;
Cory et al., 2014). These estimates suggest that the C sink strength of the land surface globally has
been overestimated, as the role of freshwater systems in the biogeochemical processing of DOC and
the subsequent production of greenhouse gases had not been considered. Understanding of the rate of
turnover of DOC in aquatic systems remains incomplete and further efforts are required to quantify
the extent to which biogeochemical processes in aquatic systems are a source of C to the atmosphere.
Photochemical reactions in aquatic systems are induced by the absorption of solar radiation,
particularly in the UV region of the spectrum, and preferentially affect aromatic, high molecular
weight (HMW) molecules derived from allochthonous sources. Upon radiation, HMW DOC is
converted to microbially available low molecular weight (LMW) carbon substrates (Opsahl and
Benner, 1998; Sulzberger and Durisch-Kaiser, 2009). Photodegradation of DOC also results in the
production of C based gases, primarily $CO_2$ and CO (Stubbins et al., 2011). Whilst it is understood
that input of photochemically labile terrigenous DOC can regulate C cycling in aquatic systems (Cory
et al., 2014; Koehler et al., 2014), the significance of DOC photodegradation processes in these cycles
remains poorly constrained over time and space (Franke et al., 2012; Moody et al., 2013). Due to low
temperatures and short residence times limiting autochthonous (in situ) DOC production in headwater



systems of northern peatlands, photochemical processing may be a proportionately more important
process.
A key control on DOC concentrations in headwater systems is rainfall events which flush young, less
degraded plant material within the catchment into streams (Evans et al., 2007; Austnes et al., 2010).
Rainfall events have been shown to contribute significantly to annual C export from peatland
headwater streams (Clark et al., 2007), yet the degree to which they replenish photolabile material
within the aquatic environment is less certain. Stormflows in northern catchments have been
associated with increased contribution of humic like material (Fellman et al., 2009), suggesting that
DOC photoreactivity may also increase during these events. Several studies have explored seasonal
variation in intrinsic DOC photoreactivity in northern aquatic systems (Vachon et al., 2016; Franke et
al., 2012) yet, to our knowledge, the contribution of rainfall events to the seasonal cycle of photolabile
material has not been previously investigated.
Further uncertainty remains in understanding the variation in DOC photolability at different positions
within a watershed (Franke et al. 2012). The increasing residence time of downstream aquatic
systems, as headwater streams drain into rivers, lacustrine and marine environments, may mean that
photo-processing becomes a more important control on overall C budgets with distance downstream.
Conversely, the extent to which the material has already been degraded in the upstream aquatic
environment may mean that further processing is limited (Catalán et al., 2016; Vähätalo and Wetzel,
2008). Investigating the susceptibility of DOC to photo-processing in different types of aquatic
environments will allow the overall contribution of photochemical processes to C cycling to be
understood on a catchment scale.
The primary aim of this study was to assess temporal variation in the photochemical lability of DOC
from two contrasting aquatic systems draining peatlands and to understand how this variation may
impact aquatic C budgets. Controlled UV irradiation experiments were conducted on water samples
collected from the two contrasting aquatic systems, one a stream and the other a reservoir. Water from
both systems was sampled on a monthly basis over a 1 year period and also from the high DOC



stream system during two rainfall events to characterise short term variability in DOC concentration
and composition. After experimental exposure, optical, spectroscopic and biogeochemical analyses of
the water samples were conducted to explore DOC photoreactivity and the resultant production of C
based gases. The results were used to test the following hypotheses:
H1: Both aquatic systems will exhibit seasonality with regards to the supply of photochemically labile
DOC, with highest photolability detected in the winter due to limited processing in the aquatic
environment.
H2: Photochemical degradation of DOC will be a more significant loss term of C in the high DOC
aquatic system.
H3: Rainfall events in the high DOC system will replenish the supply of photolabile material.
**2.   Methods**
**2.1 Study sites**
Water samples for the irradiation experiments were collected from two aquatic systems located in
peatland catchments. The Black Burn (55°47'34″ N; 3°14'35″ W; 254 m a.s.l.) is a small headwater
stream draining Auchencorth Moss, an ombrotrophic peatland located in central Scotland covering
3.35 km² (Billett et al., 2010). The stream is fed by a number of small tributaries from the surrounding
peatland, part of which is used for peat extraction. Low density sheep grazing is the primary land use
within the catchment and vegetation comprises a *Sphagnum* base layer and hummocks of
*Deschampsia flexuosa* and *Eriophorum vaginatum*, or *Juncus effusus*. In the upper catchment shrubs
are present, including *Calluna vulgaris*, *Erica tetralix* and *Vaccinium myrtillus* (Dinsmore et al. 2010;
Drewer et al., 2010).
The Black Burn stream hydrographic record is characterised by a steady base flow and rapid ('flashy')
response to rainfall events which typically produce high flow accompanied by elevated DOC
concentrations. Annual mean stream water DOC concentrations determined by weekly sampling over
a 2 year period were high, at  28.4 ± 1.07 mg L$^{-1}$ (Dinsmore et al. 2013), with a marked seasonal





pattern, characterised by low DOC in winter and high concentrations in summer. In this study, water
samples were collected from an established sampling site where DOC concentrations have been
recorded for >9 years as part of the Centre for Ecology & Hydrology (CEH) Carbon Catchments
project (https://www.ceh.ac.uk/our-science/projects/ceh-carbon-catchments).
The other sampling site was Loch Katrine (56°25'25″ N; 4°45'48″ W; 118 m a.s.l.) in the Loch
Lomond and Trossachs National Park, Scotland. Loch Katrine has a surface area of 8.9 km$^2$ and is fed
by ~88 tributaries which predominantly drain a catchment of upland blanket bog (SNH, 2005). Loch
water DOC concentrations have been recorded by the Scottish Environment Protection Agency
(SEPA) at Ruinn Dubh Aird, a peninsula located at the south eastern end of the loch, which was also
selected as the sampling point for this study. DOC concentrations measured approximately six times a
year from 2009–2014 were low at $3.68 \pm 0.56$ mg L$^{-1}$ (SEPA, personal communication).

### 2.2 Sample collection

Water was sampled monthly from both sites from May 2014 to May 2015 inclusive (13 samples over
the study duration) to characterise seasonal variation in DOC concentration and composition. Samples
were collected at 20 cm below the surface of the water in a screw top sterile clear glass bottle. Upon
return to the laboratory, samples were stored in the dark at 4°C and exposed to experimental
conditions within a week of collection. Additional water sampling to characterise the effect of rainfall
events focused on the Black Burn head water system. Intensive stream water sampling was conducted
during two rainfall events, one in winter (defined as 1 October to 31 March) and the other during the
summer (1 April to 30 September) (Gordon et al., 2004). An automatic water sampler (Teledyne Isco,
USA) was programmed to collect a composite 1 L sample of water from the Black Burn into separate
polypropylene bottles every 60 minutes (comprising two 500 mL samples collected each 30 minutes)
throughout the rainfall events. Stream water sampling in the winter rainfall event was conducted from
11:00 on 9 December to 17:00 GMT on 10 December 2014, resulting in 31 samples across the event.
Stream water sampling in the summer rainfall event started at 14:30 on 1 September and finished at
06:30 GMT on 2 September 2015, resulting in 17 samples. Water samples were transferred into glass





bottles from the automatic water sampler for transport to the laboratory and irradiated within 5 days of
collection.
Throughout the year of sampling, the Black Burn water depth was measured at 15 minute intervals
approximately 2 km downstream from the sampling site using a Level Troll pressure transducer (In
Situ Inc., USA) with atmospheric correction from a BaroTroll sensor (In situ Inc., USA) located
above the water surface. Water depth readings from the pressure transducer were converted to
discharge at the sampling site using rating curves ($R^2 > 0.90$) based on flows measured by dilution
gauging (Dinsmore et al., 2013). Equivalent hydrological data were not available for Loch Katrine.
**2.3 Sample preparation**
Prior to experiments water samples were degassed under a vacuum pressure system for 20 minutes to
remove dissolved gas from the water and then filtered using syringe driven pore size filters 0.22 μm
(Merck Millipore, UK) to exclude microbial activity. 15 mL of filtered sample was immediately
transferred into 21 mL quartz vials (Robson Scientific, UK) which were sealed with aluminium crimp
tops and rubber butyl plugs (Speck and Burke, UK). All samples were prepared at room temperature
in oxygenated conditions.
**2.4 Irradiation experiments**
Irradiation experiments were conducted using UV-B 313 lamps (Q-Panel Com, USA) covered with
125 μm cellulose diacetate (A. Warne, UK) to exclude UV-C (<280 nm) and providing both UV-A
(400-315 nm) and UV-B (315-280 nm) exposure. Lamps were mounted inside quartz tubing (Robson
Scientific, UK) beneath the water surface in a water bath maintained at 16°C and vials were irradiated
sideways while submerged. UV irradiance of the samples was modulated to remain constant
throughout the 8-h exposure by measurement with a broad-band sensor (Model PMA2102; Solar
Light Inc., USA) held beneath the water surface behind a quartz window of the same thickness as the
vials. The sensor was calibrated with a double monochromator scanning spectroradiometer
(Irradian™, UK), itself calibrated against a secondary deuterium lamp standard (FEL Lamp, F-1297)
operated by the NERC Field Spectroscopy Facility, Edinburgh (http://fsf.nerc.ac.uk/). Total





unweighted irradiance was 1.81 W m$^{-2}$ in the UV-B, 4.63 W m$^{-2}$ in the UV-A, and photosynthetically
active radiation (PAR) was 0.92 W m$^{-2}$ (Supplementary Information Figure S1). These conditions
reflect a UV-B irradiance that could be expected on a cloudless summer day in the UK and a
significant underestimation of summer time ambient UV-A and PAR radiation. The time duration of
the experiment (8 h) was selected to represent a conservative estimate of the exposure time of surface
water during transit through a headwater peatland catchment to a marine outlet. Water temperatures of
~16°C were measured in both field sites in May 2014 prior to commencement of the year-long
sampling programme and was employed in the experiments to represent summer time conditions.
Controls comprising quartz vials containing water samples and wrapped in aluminium foil to exclude
radiation were kept in the water bath for the experiment duration, with four replicates of each of the
UV-exposed and control samples.
To select water samples from the Black Burn for irradiation experiments, POC concentrations, a$_{254}$
values and E4:E6 ratios were measured within 24 h in all samples (using the methods described
below) and, from these results, eight stream water samples were selected from each rainfall event
which represented the minimum, maximum and median values of these parameters (Supplementary
Information Table S1).
**2.5 Analytical methods**
On each monthly sampling occasion the water dissolved oxygen (DO), conductivity, pH and
temperature were measured on site with a handheld Hach HQd multimeter (Hach, USA). Measured
volumes of water samples were filtered within 24 h of collection through pre-ashed (8 h at 450°C),
pre-weighed Whatman GF/F (0.7 μm pore size) filter papers. POC was determined using loss-on-
ignition, following the method of Ball (1964).
Following irradiation, partitioning of dissolved C gases from the liquid into the vial headspace was
encouraged through use of a wrist action shaker for 30 seconds. $CO_2$, $CH_4$ and CO concentrations
were measured in the vial headspace within 8 h of irradiation, using an Agilent gas chromatography
(GC) system (Agilent Technologies, USA) equipped with an autosampler and a flame ionisation



detector (FID) held at 250°C. The carrier gas was $N_2$ at a constant flow rate of 45 mL $min^{-1}$. A
methaniser fitted between the column and FID made possible $CO_2$ and CO measurements. Standard
gas mixtures (British Oxygen Company (BOC) Ltd., UK) were used for detector calibration prior to
sample analysis (detection limits were: $CO_2$ 78 ppm, CO 1.6 ppm, $CH_4$ 0.8 ppm).
DOC and dissolved inorganic carbon (DIC) concentrations were measured using a PPM LABTOC
Analyser (Pollution and Process Monitoring Ltd., UK) in UV treatment and control samples after
exposure. DIC was calculated as the difference between total carbon (TC) and DOC. UV-visible
absorbance of UV treatment and control samples contained in a 3.5 mL cuvette was measured at room
temperature between 200 and 800 nm at increments of 1 nm using a Jenway spectrophotometer
(Model 7315; Bibby Scientific, UK).  Deionised water controls were used between each sample.
Absorption coefficients $a_\lambda$ were calculated as:
$$a_\lambda = 2.303 \ x \ \left(\frac{A\lambda}{L}\right) \tag{1}$$

where A is the absorbance at each wavelength and L is the path length (m) of the cuvette (Green and
Blough, 1994). Specific UV absorbance ($SUVA_{254}$) values, a measure of DOC aromaticity, were
determined by dividing the UV absorbance measured at $\lambda = 254$ nm by the DOC concentration
(Weishaar et al., 2003). E4:E6 ratios were estimated using the absorbance values at 465 and 665 nm,
respectively (Peacock et al., 2014).
Fluorescence intensity in water samples filtered to 0.2 μm was measured using a FluroMax-4
spectrofluorometer (Horiba Jobin Yvon Ltd., Japan). The instrument was programmed to scan across
excitation wavelengths 200-400 nm (5 nm increments) and emission wavelengths 250-500 nm (2 nm
increments) with a 1 nm path interval. Data were obtained at room temperature and were blank
corrected using deionised water. Intensity ratios derived using these data allow discrimination
between different sources of DOC. Here, the fluorescence index (FI), $f_{450}/f_{500}$, the ratio of fluorescence
intensity at the emission wavelength 450 nm to that at 500 nm at excitation wavelength 370 nm, was
calculated to help identify dissolved organic matter (DOM) source material. Values around 1.8





suggest autochthonous organic material, whereas values around 1.2 indicate terrestrially derived
material (Cory and McKnight, 2005).
Lignin phenol concentrations in unirradiated Black Burn water samples were measured using the CuO
oxidation method (Benner et al., 2005; Spencer et al., 2008). After filtration to 0.2 μm, 45 mL of
water sample was freeze dried to produce lyophilised DOM which was transferred to stainless steel
pressure bombs with 1 g of CuO and 100 mg of $Fe(NH_4)_2(SO_4)_2H_2O$. Under anaerobic conditions, 8
mL of NaOH was added to the bombs before they were sealed. Samples were then oxidised at 155°C
for 3 h. Following oxidation, samples were acidified to pH 1 with $H_2SO_4$, extracted with ethyl acetate
three times, and then passed through $Na_2SO_4$ drying columns. Samples were dried using a flow of $N_2$
and kept frozen prior to GC analysis. After redissolution in ~200 μL pyridine, lignin phenols were
derivatised with bis-trimethylsilyltri-fluoromethylacetamide (BSTFA) and quantified on a GC
(Agilent 5890 MkII with twin FID).
Eleven lignin phenols were measured, including three p-hydroxybenzene phenols (P): p-
hydroxybenzaldehyde, p hydroxyacetophenone, p-hydroxybenzoic acid; three vanillyl phenols (V):
vanillin, acetovanillone, vanillic acid; three syringyl phenols (S): syringaldehyde, acetosyringone,
syringic acid; and two cinnamyl phenols (C): p-coumaric acid and ferulic acid. Blank controls, taken
through the method from CuO oxidation onwards, were quantified and subtracted from sample
concentrations. Quantification was achieved through use of cinnamic acid as an internal standard. In
addition to total concentration of lignin phenols ($\Sigma_{11}$) and carbon normalised yields ($\Lambda_{11}$), the ratio of
syringyl to vanillyl phenols (S/V), the ratio of cinnamyl to vanillyl (C/V) phenols, the ratio of p-
hydroxybenzenes to vanillyl phenols (P/V) and the ratio of acids to aldehydes ($Ad/Al_{v,s}$) were
calculated to aid interpretation of the data. Lignin phenols for Loch Katrine samples were not
measured due to insufficient production of lyophilised material using the stated method.
**2.6 Data analysis**
Data collected in the irradiation experiments were tested for normality using the Shapiro-Wilks test
and were found to be normally distributed. Unpaired t-tests were conducted between irradiated and



unirradiated samples to assess differences in spectral properties, DOC and DIC concentrations, lignin
phenol concentration and gaseous production. Pearson correlation coefficients were used to test the
potential role of DOC composition and site conditions in regulating photochemical lability, measured
as total DOC loss, production of DIC and C gases (CO and $CO_2$) and change to $a_{254}$ and E4:E6 ratios.
Carbon species DOC, DIC, $CO_2$ and CO measured each month at the Black Burn and Loch Katrine
were included in C mass budgets calculated for irradiated and unirradiated samples. By converting all
data to mg $L^{-1}$, the difference in C budget between treatment and control samples could be determined
(see Supplementary Information Table S2 for example calculations). To obtain a standard error value
for differences between irradiated and control samples, the mean control value was determined and
subtracted from each of the irradiated replicates.
Correlation coefficients were also calculated between intrinsic sample photoreactivity, measured as
total change to C species upon irradiation normalised for initial DOC concentration, and lignin phenol
data. The Durbin-Watson statistic was used to test for the presence of autocorrelation in residuals of
lignin phenol analyses of stream water samples collected during rainfall events and showed no
correlation between the samples. Minitab v.16 (Minitab Inc., USA) was used for all statistical
analyses.
**3.  Results**
**3.1 Climate and water chemistry conditions at time of sampling**
Total rainfall measured at the European Monitoring and Evaluation Programme (EMEP) supersite at
Auchencorth Moss (Torseth et al., 2012) for the 13 month sampling period was 1015 mm. It varied
from lowest monthly values in September and April to the highest in October (Figure 1a). The mean
air temperature of the study period was 7.7°C, similar to the 8 year average of 7.6°C, and reached a
maximum of 27.6°C in July 2014 and a minimum of -7.9°C in January 2015.
At Comer meteorological station, located 10 km from the Loch Katrine sampling site, rainfall was
considerably higher, totalling 2368 mm over the sampling period (Figure 1b) (Met Office, 2012).

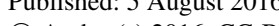
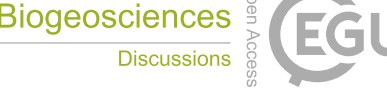


Seasonal variation in rainfall was clear, with >40 % of rainfall falling from December to February.
Air temperatures were higher than at the Black Burn, with a mean of 10.2°C.
Water chemistry differed considerably between the two aquatic systems over the year-long sampling
(Table 1). The water temperatures reflected the difference in air temperature between the sites, with
higher mean values at Loch Katrine than at the Black Burn. Mean pH at the Black Burn was 5.4,
compared to 6.7 at Loch Katrine. Conductivity was more variable at the Black Burn and was on
average 53 µS cm$^{-1}$ higher than at Loch Katrine, although values at both sites were low. POC
concentrations at the Black Burn were over double those at Loch Katrine. FI values were slightly
higher at the Black Burn, but at both sites were low and stable, indicative of terrestrially derived DOC
material (Cory and McKnight, 2005)
DOC concentrations at the Black Burn ranged from 14.2 to 50.9 mg L$^{-1}$ (Figure 2) and showed a
similar seasonal pattern as described in Dinsmore et al. (2013). Concentrations were lowest in late
winter and highest in autumn; the latter consistent with increased organic matter inputs to the stream
from flushing of soils during autumn rainfall events.
At Loch Katrine, DOC concentrations were low and consistent, ranging from 3.10 to 5.82 mg L$^{-1}$.
Concentrations were lowest in spring and highest in summer. SUVA$_{254}$ values at the Black Burn were
higher than at Loch Katrine, suggesting that the DOC pool was comprised of a greater percentage of
aromatic material (Weishaar et al., 2003). The E4:E6 ratio at the Black Burn varied considerably over
the sampling period, ranging from 1.0 to 10.2. At Loch Katrine, the E4:E6 ratios were lower and less
variable, but are a less meaningful parameter in the low DOC concentration Loch Katrine samples due
to minimal absorbance in wavelengths greater than 400 nm.
**3.2 Optical changes in water samples upon irradiation**
Absorbance coefficients typically decreased upon irradiation of water samples, with the strongest
decrease occurring in the UV part of the spectrum at ~225 nm, and a smaller inflection at ~300 nm
(Figure 3). The maximum change in absorbance upon irradiation was a factor of 4 higher in water
samples from the Black Burn than from Loch Katrine. In the Black Burn, decreases in absorbance



were greater in the summer and autumn, whereas at Loch Katrine the decreases in absorbance were
greater in the winter and spring.
Positive values (where dark control samples showed a greater drop in absorbance upon irradiation
than light exposed samples) were recorded for summer water samples from Loch Katrine. E4:E6
ratios decreased by a mean of 1.52 in irradiated Black Burn water samples, indicating accumulation of
increasingly humic material in the remaining DOC pool during light exposure. At Loch Katrine,
E4:E6 ratios decreased by a mean of 0.21 upon irradiation.
**3.3 Carbon budget changes upon irradiation**
Typically, DOC concentrations in Black Burn water samples decreased after light exposure compared
to unirradiated controls (Figure 4a). Mean change in DOC in irradiated samples from the Black Burn
for the whole sampling period was -2.14 mg C $L^{-1}$ (ranging from 0.06 to -4.35 mg C $L^{-1}$ for individual
months). DOC decreased after irradiation in all Black Burn samples with the exception of September
2014, indicating a photolabile DOC pool for most of the year. In contrast, in water samples from Loch
Katrine irradiation induced DOC losses occurred in 6 of 13 samples and small gains were observed in
7 of 13 samples (Figure 4b). Whilst these results should be interpreted with caution as small
differences in DOC concentrations (<0.5 mg C $L^{-1}$) are below the instrument detection limit, they
suggest that the DOC pool in Loch Katrine was largely recalcitrant to photochemical degradation.
Irradiation resulted in notable photoproduction of DIC, $CO_2$ and CO from Black Burn samples. DIC
concentration increased by a mean of 0.77 mg C $L^{-1}$ for the whole sampling period, although
production across the samples was highly variable between months. $CO_2$ was the most abundant
photoproduct and was produced at a mean rate of 1.2 mg C $L^{-1}$ across all monthly samples. At Loch
Katrine, $CO_2$ production was two orders of magnitude lower than in the Black Burn, produced at a
mean rate of 0.06 mg C $L^{-1}$. In all monthly water samples from both sites CO concentrations increased
in the irradiation experiments, with mean production rates of 0.07 and 0.01 mg C $L^{-1}$ observed for
Black Burn and Loch Katrine samples, respectively.



Carbon mass budgets for DOC loss and photoproduct accumulation (DIC, $CO_2$ and CO) in water
samples were calculated for all the irradiation experiments. Budgets for all monthly water samples
from the Black Burn were balanced to within ± 5.1% of the total measured C concentration. For Loch
Katrine water samples, budgets were balanced to within ± 11%. The lower accuracy of budget closure
in the Loch Katrine samples is likely due to lower overall C concentrations, which are more
susceptible to measurement error. $CH_4$ was detected in all samples at very low levels, with mean
concentrations of 0.63 and 0.57 µg $L^{-1}$ detected at the Black Burn and Loch Katrine, respectively, and
thus were not included in the mass calculations.
Intrinsic photoreactivity of C in the Black Burn ranged from 0.02 to 0.15 mg C/mg DOC $L^{-1}$ and was
highest in August (Figure 4a). Photoreactivity peaked again in November and remained elevated until
January. Lowest sample photoreactivity was detected in September. At Loch Katrine, mean C
photoreactivity was 0.004 mg C/mg DOC $L^{-1}$, with a maximum of 0.09 mg C/mg DOC $L^{-1}$ detected in
July.
**3.4 Factors influencing carbon budget changes**
Factors influencing irradiation induced changes to C species and spectral properties in Black Burn
water samples were investigated using Pearson correlations (Table 2). Loss of DOC, absorbance at
254 nm and production of both $CO_2$ and CO were significantly positively correlated with initial DOC
concentration. Initial E4:E6 ratios had positive coefficient values with all light induced changes to the
DOM pool, whilst FI values were all negative, although most of these correlations were not
significant.
Of the meteorological and discharge variables investigated, air temperature and PAR were
significantly negatively correlated with changes to E4:E6 ratios. Total monthly rainfall had positive
coefficient values with irradiation induced changes to the DOM pool. Correlations between C species
changes and discharge were less consistent, although mean monthly discharge was significantly
positively correlated with changes to E4:E6 ratios.





### 3.5 Effect of rainfall events on carbon photo-processing in Black Burn water samples

The Black Burn was sampled hourly during a winter rainfall event, with collection commencing 6 h before peak rainfall (Figure 5a). Total rainfall during the event, which we define here as the water sampling period, was 19.6 mm, with an hourly maximum of 3.3 mm and rainfall recorded in 22 of the 31 sampling hours. Stream discharge peaked at 391 L s$^{-1}$ although a separate smaller peak of 266 L s$^{-1}$ also occurred during the sampling period.

During the event, an initial dilution of stream DOC concentrations was followed by recovery to pre-event levels (Figure 5a). DOC was most photoreactive at 06:00, with DOC concentration reduced after irradiation by 6.72 mg L$^{-1}$. DOC loss in this sample was greater than at any time through the year-long study (Figure 4a), even though the DOC concentration (44.4 mg L$^{-1}$) was within the range of measured monthly concentrations. The greatest irradiation induced increase in $CO_2$ concentration (2.25 mg L$^{-1}$) occurred in the first event sample at 11:00, collected prior to rainfall input. Photoreactivity was lowest at 12:00, and was similarly low in the sample collected at 17:00, which coincided with peak rainfall.

In the late summer rainfall event occurring at the end of an extended period of base flow in the Black Burn (Supplementary Information Figure S2), 3.2 mm of rainfall was recorded with a maximum hourly total of 2.2 mm. Samples were collected from 14:30 to 06:30, with rainfall only occurring between 16:30 and 18:30. Discharge remained low and relatively stable throughout the event, with a mean flow of 6.14 L s$^{-1}$. Rainfall marginally diluted the stream DOC concentrations (Figure 5b). Photo-induced changes were much smaller than in the winter event and maximum DOC losses were a factor of 2.5 lower than the mean DOC reduction observed in the Black Burn monthly water sample experiments (Figure 4a). Photoreactivity was lowest in the initial sample collected at 14:30 prior to rainfall and coinciding with the highest discharge during the sampling period. Photoreactivity was highest in the 19:30 sample collected 3 h after peak rainfall.





### 3.6 Lignin phenol composition of Black Burn water samples

To understand the effect of DOM composition on photolability, lignin phenols were measured in all

the Black Burn monthly and rainfall events water samples prior to the irradiation experiments.

Dissolved lignin concentrations ranged from 15.3 to 108 µg L$^{-1}$ (mean = 52.8; n = 28) and were

significantly positively correlated with sample DOC concentration (Pearson = 0.831; p < 0.01)

(Supplementary Information Figure S3). Carbon normalised yields were between 0.71 and 2.66 mg

(100 mg OC)$^{-1}$. The contribution of individual phenol groups to the lignin signature varied between

monthly samples of the year-long study and the rainfall events (Figure 6). In the monthly samples, the

P phenols were most abundant, followed by V phenols (Figure 6a). Samples in the winter rainfall

event contained higher and more variable mean yields for each phenol group, with S phenols most

abundant, followed by V phenols and P phenols.

Overall yields were significantly lower (1-way ANOVA, p < 0.01) during the summer rainfall event.

As in the year-long samples, P phenols were the most abundant, followed by S phenols and V

phenols. Across all three sampling regimes, the contribution of C phenols to the overall lignin

signature was smallest.

P:V ratios, an indication of *Sphagnum* derived DOC (see section 4.2), ranged from 0.83 to 1.69 across

all samples, indicating significant temporal variability in DOM source material. Photoreactivity was

significantly negatively correlated with P:V ratios when all samples were combined in a correlation

analysis (-0.523; p < 0.01) (Figure 7a). This suggests that the relative abundance of P versus V

phenols contributed considerably to sample photoreactivity. The lowest P:V ratios were in winter

rainfall event samples, where photoreactivity was highest.

Ad:Al$_{v,s}$ ratios, which are an indicator of sample degradation, ranged from 0.58 to 1.26, towards the

lower end of reported values in the literature (Winterfeld et al., 2015). Photoreactivity was also

significantly negatively correlated with Ad:Al ratios (-0.492; p < 0.01) (Figure 7b) and again lower

ratios typically occurred in winter rainfall event samples.





## 4. Discussion

### 4.1 Peatlands as a source of photochemically labile DOC

Photo-processing resulted in considerable DOC loss from water samples from the Black Burn. Mean DOC loss in the 8 h irradiation experiments conducted on the monthly water samples was 6% relative to initial concentrations. Percentage DOC losses determined here are similar to those reported from irradiation experiments conducted over similar timescales using stream water draining a boreal watershed (3–10 % DOC loss over 10 h; Franke et al., 2012 and 11% TOC loss over 19 h; Köhler et al., 2002). Photochemical transformations were low in the Loch Katrine samples, with minimal losses to the DOC pool (-0.03%; mean from year-long study). Whilst our sites were not located within the same watershed, it seems likely that position within the catchment plays a role in determining the photolability of DOC. The Black Burn headwater stream at Auchencorth Moss receives fresh inputs of DOC from the surrounding peatland catchment and material has less time for light exposure in the water column relative to the DOC in the reservoir system. DOC losses may occur in Loch Katrine soon after water entry into the loch but, due to long water residence times, DOC may have become recalcitrant to photo-processing by the time of sample collection. Catalán et al. (2016) observed a negative relationship between organic carbon decay and water retention time, resulting in decreased organic carbon reactivity along the continuum of inland waters. $SUVA_{254}$ data suggest that DOC in Loch Katrine samples was less aromatic than in the Black Burn (Table 1), with values indicating an approximate humic content of 30% based on the findings of Weishaar et al. (2003). As humic molecules are more labile to photo-processing, irradiation had a greater effect on the stream samples relative to the reservoir samples.

Strong seasonal fluctuations in DOC concentration and composition occurred in the Black Burn, in agreement with patterns observed in the same system by Dinsmore et al. (2013). DOC concentrations were highest in the late autumn, consistent with a flushing effect whereby soil organic material produced over the summer is mobilised and delivered to aquatic environments by more intense rainfall after a prolonged, relatively dry period (Fenner et al., 2005). Positive correlation between the irradiation induced change in the E4:E6 ratio and mean monthly discharge suggest that hydrological





conditions in the month prior to sampling significantly influence the reactivity of the sample, with
high flow delivering more reactive carbon to the stream. Overall the magnitude of photo-induced C
losses was significantly positively correlated with DOC concentration in the year-long Black Burn
dataset. However, despite low DOC concentrations, photoreactivity remained elevated in January.
This suggests that even when lower DOC concentrations are detected in aquatic systems, the DOC
may be intrinsically more photoreactive due to its aromatic content and minimal light exposure
history.
Lowest DOC concentrations were observed in the late winter and early spring, due to depletion of soil
organic C within the catchment by autumn and winter rainfall events. Low rainfall inputs limit the
recharge of fresh, photolabile material to the stream and may account for the reduction in DOC
photoreactivity detected in September. Furthermore, due to longer residence time in the water column,
these samples may have already been degraded by natural light. A previous study at the Black Burn
reported $^{13}$C enrichment of stream water DOC in September, consistent with increased in-stream
processing at this time of year (Leith et al., 2014). Reductions in intrinsic DOC photolability during
summer have similarly been reported in northern lakes (Vachon et al., 2016) and a boreal watershed
(Franke et al., 2012).  Another minimum in photoreactivity occurred in April, where SUVA$_{254}$ data
indicate decreased contribution of aromatic material to C within the stream. Although algal abundance
was not measured during this study, production of DOC from such sources would account for the
reduction in photolability (Nyugen et al., 2005).
Whilst DOC losses from Loch Katrine water samples were minimal, the peak in photolability,
indicated by the greatest absorbance reduction in the light exposure experiments, occurred in spring.
Similar seasonal photolability peaks have been observed in northern lakes (Vachon et al., 2016) and
boreal streams (Porcal et al., 2013) and are partly attributed to mobilisation of terrigenous material
with high flows associated with spring snow melt. The magnitude of melt in the Loch Katrine
catchment will be considerably less than in snow dominated northern catchments (e.g. Laudon et al.,
2013), although increased flow and stream water chemistry changes with spring snow melt have been
reported in upland Scottish catchments (Abrahams et al., 1989; Gilvear et al., 2002).

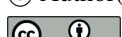



Absorbance increased in light exposed samples during irradiation in summer Loch Katrine samples,
indicating production of DOC. Prior filtration of samples to 0.22 µm means that this effect is unlikely
to be the result of microbial DOC production. A possible explanation for increased absorbance in the
irradiated water samples is the formation of an iron (Fe)-DOC complex, since the reaction kinetics of
Fe-DOC complexes are directly affected by light exposure (Maranger and Pullin, 2003). Whilst Fe
concentrations were not measured in this study, in a long term SEPA bimonthly measurement
campaign (2009-2013) at Loch Katrine, peak Fe concentrations in August of up to 0.50 mg L$^{-1}$ were
detected, corresponding to the time of year when we found increased absorbance in water samples. As
the data set does not cover the sampling period, the role of Fe-DOC complexes in producing the
observed effect cannot be directly determined; however the role of micronutrients in peatland aquatic
C cycling should be further investigated.
**4.2 Importance of rainfall events in mobilising photolabile material**
Dissolved lignin phenol composition indicates that different sources of plant material were mobilised
as a result of rainfall in the Auchencorth Moss catchment. High P:V ratios have been used as an
indicator of peatland inputs to aquatic systems, as *Sphagnum* acid typical of peatlands is converted
into P phenols during lignin extraction (Fichot et al., 2016; Winterfeld et al., 2015). Typically P
phenols constituted the largest contribution to the total lignin concentration of the measured phenols,
consistent with *Sphagnum* inputs. However, during the winter rainfall event where stream discharge
was considerably higher than the year-long mean value, the largest contribution to total lignin
concentration was from S and V phenols (Figure 6). The former are reported to be the most
photolabile phenol (Opsahl and Benner, 1998) and are unique to woody angiosperms. This suggests
that hydrological pathways within the catchment were activated upon rainfall, causing DOC release
from soil profiles associated with angiosperm plant material. Potential sources within the Auchencorth
upper catchment are *Calluna vulgaris*, *Erica tetralix* and *Vaccinium myrtillus*. Further evidence of the
operation of variable source areas in the catchment was the observation of delayed input of water,
containing high $CO_2$ concentrations, from the deep peat area in the upper catchment at Auchencorth
Moss during a storm event (Dinsmore and Billett, 2008). Low P:V values and high lignin



concentrations have been reported during peak flow in Arctic rivers, and the reverse during base flow
(Amon et al., 2012). As samples with low P:V values were typically more photoreactive (Figure 7a),
our data indicate that rainfall events are important in mobilising photolabile material from this
catchment.
Elevated Ad:Al$_{v,s}$ ratios have previously been interpreted as indicators of decomposition of organic
matter resulting from preferential degradation of aldehydes relative to acids (Spencer et al., 2009). In
the Black Burn water samples, lowest ratios were measured in the winter rainfall event. This implies
that DOC mobilised during rainfall is less degraded relative to base flow DOC, in agreement with
previous studies of peatland high flow events which detected increased contribution of near surface
flow and younger DOC (Clark et al., 2008). The form of the degradation, either microbial or
photochemical, cannot be distinguished using these data. However, based on the higher measured
photoreactivity of samples with lower ratios (Figure 7b), light exposure history may be one of the key
moderators of Ad:Al$_{v,s}$ ratios in the Black Burn. High flow events release fresh DOC from soils
derived from recent plant material (Evans et al., 2007) and may have significant implications for C
processing rates in streams as they are recharged with labile material (Lapierre et al., 2013).
Whilst the samples collected during the winter rainfall event were clearly distinct in composition
relative to samples from the year-long study, the summer rainfall event samples had similar P:V and
Ad:Al$_{v,s}$ ratios, but significantly lower photoreactivity and overall lignin yields (Figures 5b, 6c, 7).
This could be attributed to the timing of sample collection in early September at the end of summer,
where considerable degradation may have already occurred across all phenol groups so that the DOC
pool remaining was more recalcitrant to further photo-processing. Discharge data indicate that there
was no discernible flushing effect during the summer rainfall event, with slight decreases in DOC
concentration attributed to dilution of the stream water by direct rainfall inputs or overland flow. The
abundance of P phenols within the samples suggest that passive transfer of DOC from the riparian
zone, which is dominated by *Sphagnum* and *Juncus* vegetation, to the stream was the dominant mode
of stream DOC recharge at this time of year (Jeanneau et al., 2015). The summer rainfall event



samples were notably depleted in V phenols, suggesting that these phenols exert an important control
on sample photoreactivity in addition to S phenols.

**4.3 Implications for photochemical turnover of DOC in aquatic systems**

DOC loss from samples upon irradiation resulted in significant production of $CO_2$. The mass budget
calculations for Black Burn water samples show that a mean of ~46% of DOC loss in the irradiation
experiments was accounted for by production of $CO_2$. Dinsmore et al. (2010) estimate that $108 \pm 62.7$
kg DOC $yr^{-1}$ is exported to the Black Burn from the Auchencorth Moss catchment. Based on our
finding that 7% of DOC is removed via photo-processing, and assuming that 46% of this loss is
converted to $CO_2$ and also that UV-B irradiance was comparable to a clear sky summer day, we
estimate a potential evasion loss of $3.48 \pm 2.02$ kg $CO_2$ $yr^{-1}$ to the atmosphere. Whilst this calculation
makes significant assumptions in upscaling from 8 h exposure experiments, it highlights the potential
importance of photo-processing in the turnover of aquatic C and the need for more in situ studies.
Due to the effects of bank shading and short transit time of water within the immediate catchment,
light driven instream DOC processing is unlikely to be significant. The river continuum concept
suggests that increased DOC processing will occur further downstream, where the channel widens
(Vannote et al., 1980), and will be partly controlled by the stream water mean transit time (McDonnell
et al., 2010; McGuire and McDonnell, 2006). Based on mean velocity (~0.58 m $s^{-1}$) of a larger nearby
river ( Ledger, 1981), we estimate a mean water transit time of 19 h from the Black Burn at
Auchencorth Moss to its coastal outlet in the River Esk 34 km downstream, considerably longer than
the exposure time in our experiments. However, in a study of $1^{st}$ to $4^{th}$ order streams in Sweden no
significant change to DOM composition as stream order increased was detected and this was partly
attributed to short transit times (<2 days) restricting DOC processing (Kothawala et al. 2015).
Peatland derived carbon in this study is clearly photoreactive, but limited time for in-stream
processing may render photo-processing unimportant in freshwater aquatic C budgets.
Determining the C cycling implications of this study is further complicated as the most photoreactive
material was recorded during a heavy winter rainfall event. The potential for photochemical



transformation of DOC within the freshwater aquatic environment would have been limited due to
low light availability, extensive cloud cover and increased stream water transit times associated with
the event. During the year-long study period, 12 rainfall events occurred which resulted in similar
flow conditions in the Black Burn (stream discharge exceeding 250 L s$^{-1}$), with a maximum discharge
of 2059 L s$^{-1}$ in a late winter storm. Of these high flow events, 11 occurred during winter and one in
summer and hence, whilst large quantities of photoreactive material may have been mobilised during
heavy rainfall, the likelihood of in-stream processing would remain small. Increases in precipitation,
with more frequent and intense rainfall events, are expected with climate change (Capell et al., 2013;
Edenhofer et al., 2014) with heavier summer downpours predicted in the UK (Kendon et al., 2014).
Thus, although the contribution of rainfall events to freshwater aquatic C cycling in this study is likely
to be minimal, they could become more significant if heavy rainfall events occur more frequently in
summer.
**Author Contributions**
AEP collected field samples and undertook laboratory analyses. Data analysis and writing of the paper
were also carried out by AEP. KVH, ARM and KJD provided guidance on the scope and design of the
project, and contributed to the editing of the manuscript.
**Acknowledgements**
This work was funded by a Natural Environment Research Council (NERC) PhD studentship
(NE/K500835/1). Further support was provided by a Moss PhD scholarship courtesy of Derek and
Maureen Moss. The Irradian™ spectroradiometer used in this study was calibrated by Chris McLellan
at the NERC Field Spectroscopy Facility. We thank Stephen Mowbray for his assistance with lignin
phenol analyses and Andrew Addison for his contribution to fieldwork. We also thank Tony
Dickinson and Jim Donnelly at the University of Central Lancashire for use of a Horiba FluroMax-4
spectrofluorometer.

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



Table 1. Mean (n=13 ± 1 standard deviation) water temperature and chemistry parameters including pH,
conductivity, POC concentrations, and FI values at the Black Burn and Loch Katrine.

|  | **Black Burn** | **Loch Katrine** |
|---|---|---|
| Water temperature °C | 8.26 ± 4.53 | 10.9 ± 5.07 |
| pH | 5.38 ± 0.85 | 6.74 ± 0.32 |
| Conductivity μS cm$^{-1}$ | 78.2 ± 30.7 | 25.2 ± 4.01 |
| POC mg L$^{-1}$ | 5.78 ± 2.78 | 2.96 ± 0.63 |
| FI value | 1.15 ± 0.13 | 1.08 ± 0.18 |






















Table 2. Pearson correlation coefficients between irradiation induced changes to aqueous carbon species and
spectral properties, and water chemistry of Black Burn water samples from the year-long sampling campaign
prior to irradiation and site conditions at Auchencorth Moss (n=13).

| | $\Delta$DOC | $\Delta$DIC | $\Delta$CO$_2$ | $\Delta$CO | $\Delta a_{254}$ | $\Delta$E4:E6 |
|---|---|---|---|---|---|---|
| **DOC** | **0.708**** | -0.074 | **0.773**** | **0.824**** | **0.766**** | 0.095 |
| **E4:E6** | 0.366 | 0.049 | 0.463 | 0.434 | 0.183 | **0.770**** |
| **SUVA$_{254}$** | 0.228 | 0.460 | 0.232 | 0.129 | 0.231 | -0.098 |
| **FI** | -0.438 | -0.161 | -0.318 | -0.238 | -0.115 | -0.485 |
| **Air temperature[a]** | -0.032 | -0.379 | -0.029 | -0.052 | 0.220 | **-0.571*** |
| **Rainfall[b]** | **0.603*** | 0.061 | 0.537 | 0.445 | 0.365 | 0.492 |
| **PAR[c]** | -0.161 | -0.459 | -0.380 | -0.267 | -0.224 | **-0.662*** |
| **Discharge[d]** | 0.132 | 0.237 | 0.123 | 0.088 | -0.139 | **0.767**** |

\* $p < 0.05$
\*\* $p < 0.01$
[a] Mean monthly air temperature
[b] Total monthly rainfall (mm)
[c] Mean monthly PAR ($\mu$mol m$^{-1}$ s$^{-1}$)
[d] Mean monthly discharge (L s$^{-1}$)















Figure 1. Mean monthly air temperature, total rainfall and mean discharge from May 2014 to May 2015 are
shown for a) Auchencorth Moss, with discharge of the Black Burn shown on the left hand offset axis. Mean
monthly air temperature and total rainfall are shown for the same period for Comer meteorological station, near
Loch Katrine. Note inverted right hand y axes.

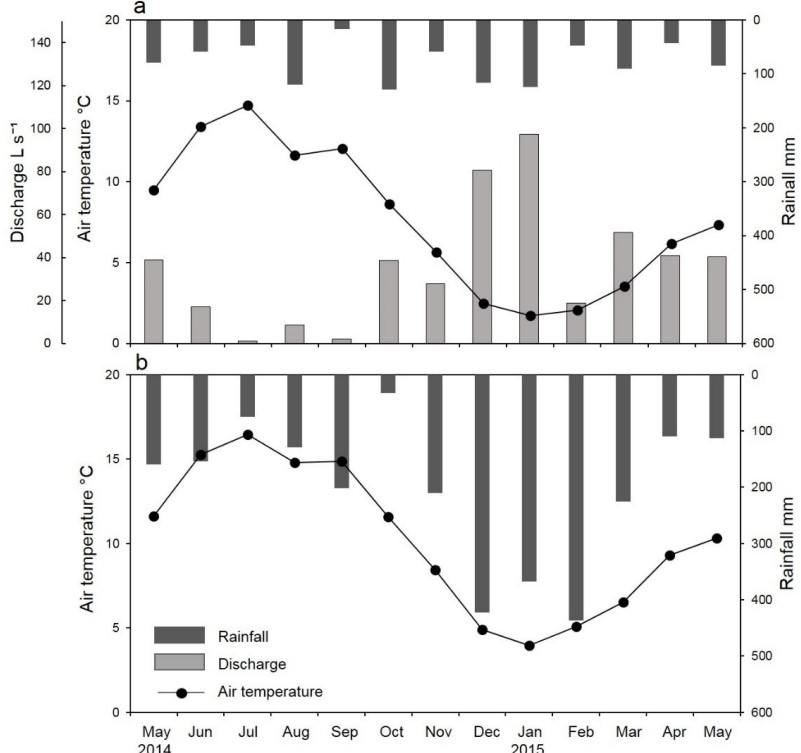












Figure 2. Time series at a) the Black Burn and b) Loch Katrine of DOC concentration and parameters for DOC
quality: SUVA$_{254}$ and E4:E6 from May 2014 to May 2015. Note different y axis scales for DOC data.

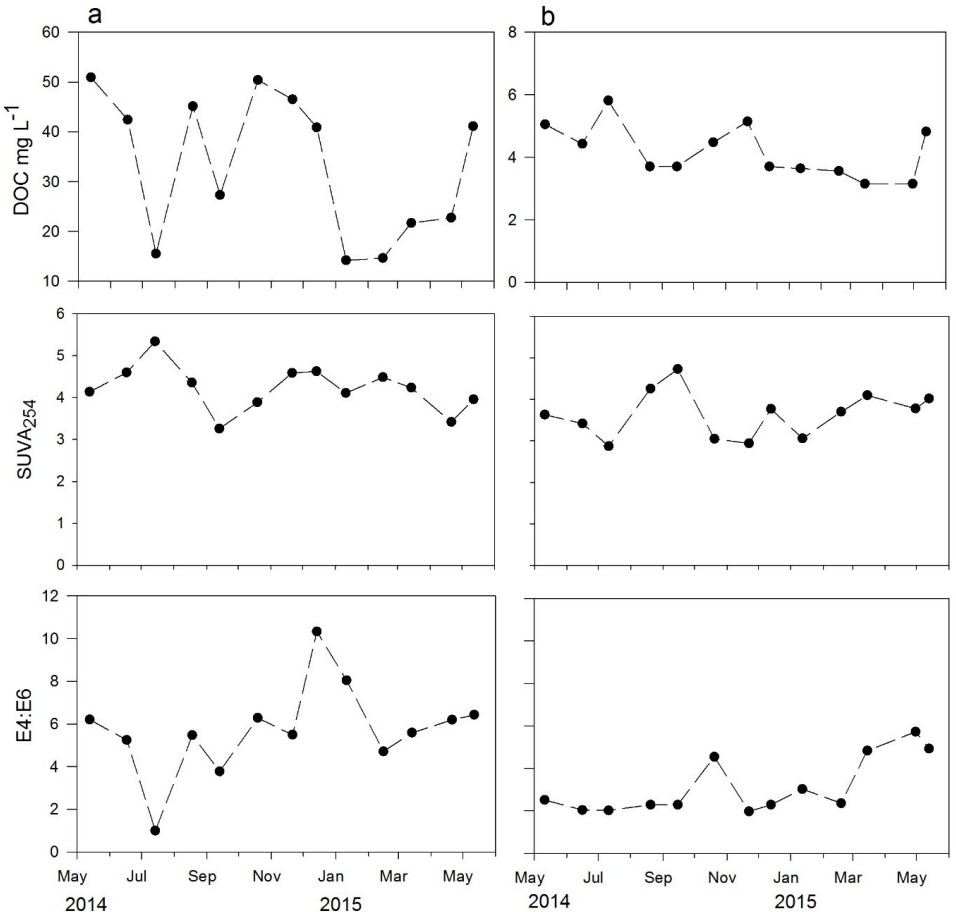













Figure 3. Irradiation induced changes (light exposed subtracted from dark controls) to water sample absorbance
values at a) Black Burn and b) Loch Katrine. Summer is the mean of June, July and August values, autumn is
the mean of September, October and November values, winter is the mean of December, January and February
values and spring is the mean of March, April and the combined mean of May '14 and May '15 values. Note
different y axis scales.

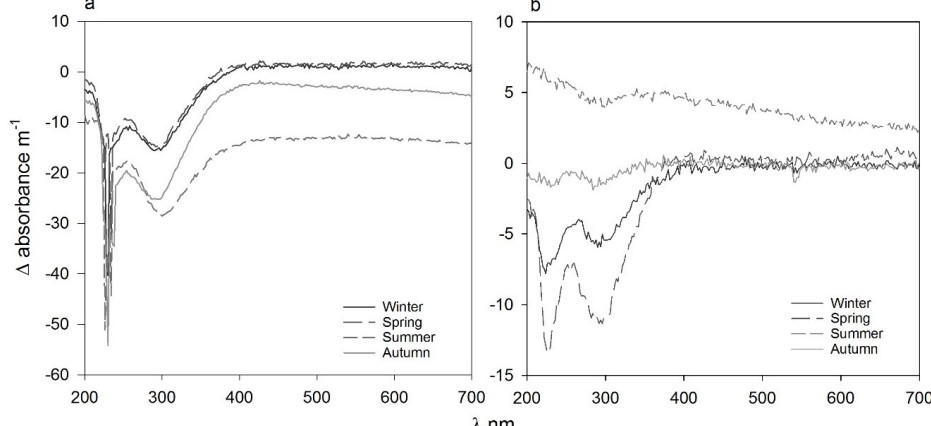
















Figure 4. Irradiation induced changes to carbon species DOC, DIC, $CO_2$ and CO in monthly water samples from
panel Black Burn (panel a) and Loch Katrine (panel b). DOC normalised changes to all C species changes are
shown on the bottom row. Data represent the difference between the mean of irradiated and unirradiated control
samples. Error bars show the standard error of the mean (n=4).

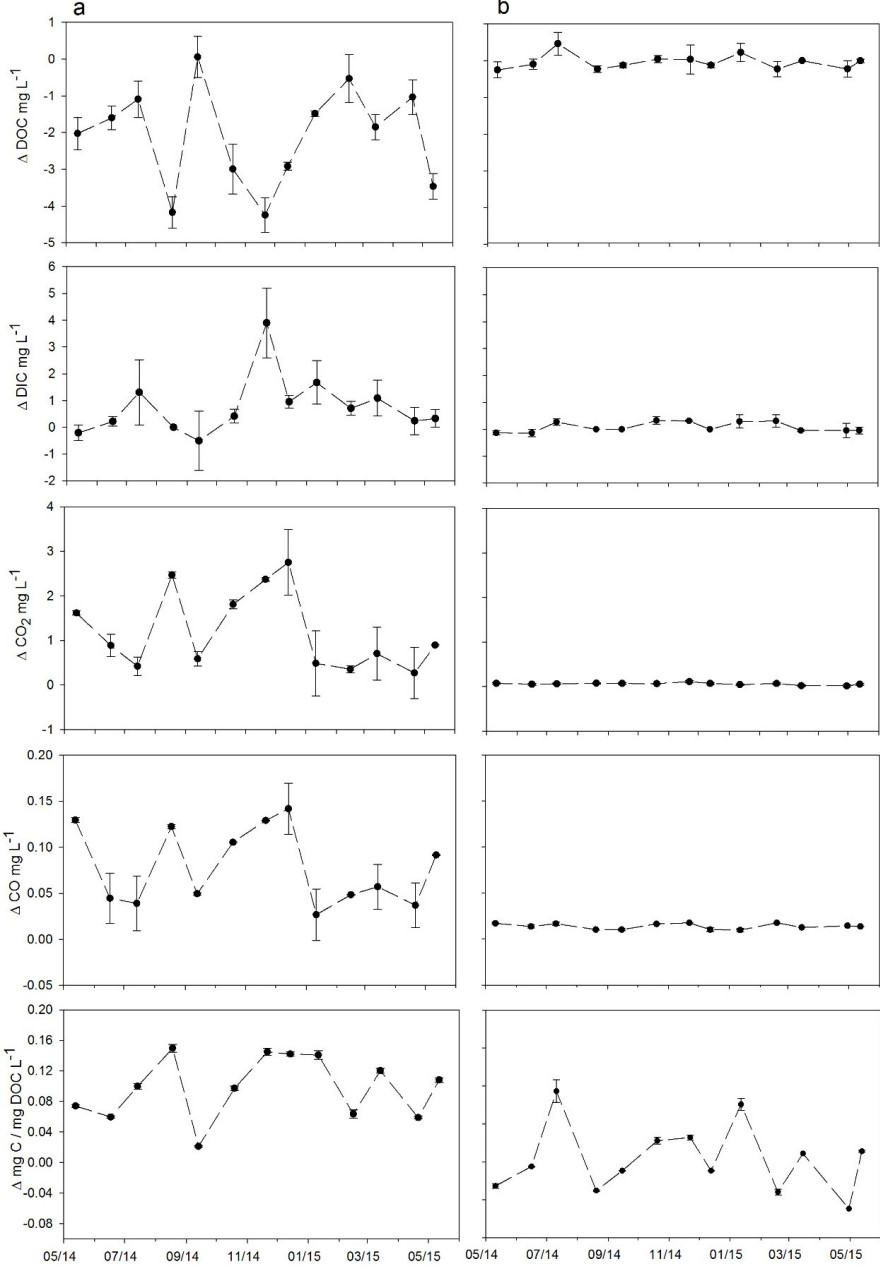






Figure 5. Rainfall events sampled on 9-10 December 2014 (panel a) and on 1-2 September 2015 (panel b). Row
one shows a time series of hourly rainfall, discharge and DOC concentrations for each event. Row two shows
photo-induced C pool changes of irradiated samples expressed as a total change value per C species in vertical
bars (left y axis) and as a DOC normalised value in dots (right y axis). Data represent the difference between the
mean of irradiated and unirradiated control samples (n=4). Note different x- and y-axis scales.

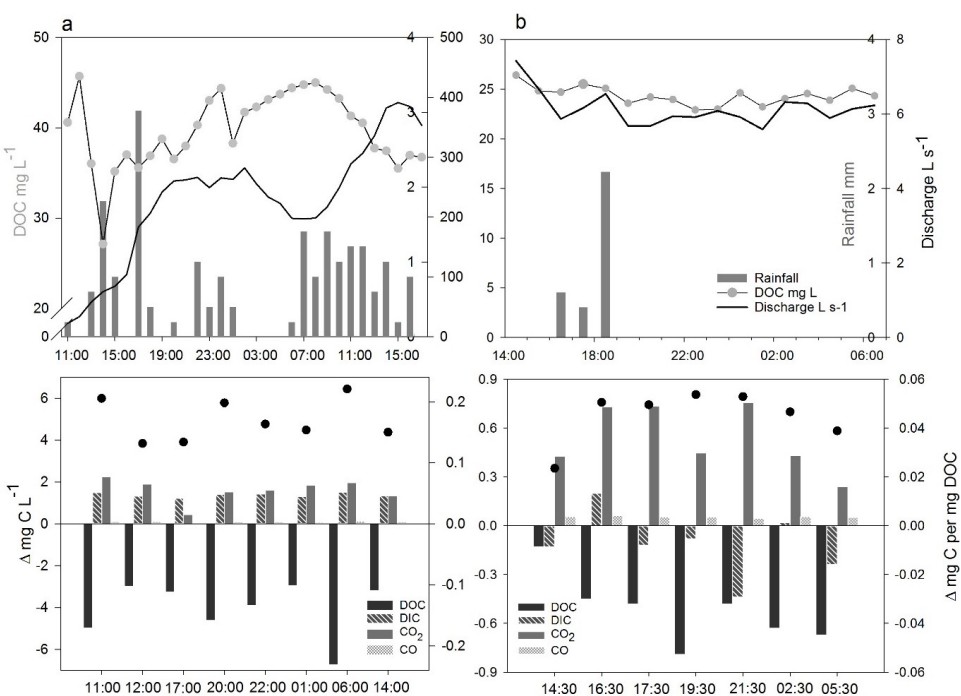















Figure 6. Boxplots of carbon-normalised yields of phenols groups for Black Burn water samples collected a)
monthly in the year-long study (n=13), b) during the winter rainfall event (n=8) and c) during the summer
rainfall event (n=7). P = p hydroxyl, V = vanillyl, S = syringyl and C = cinnamyl. The box spans from the first
quartile to the third quartile, with the line showing the median value. Whiskers show the minimum and
maximum values, with dots representing outlying values.

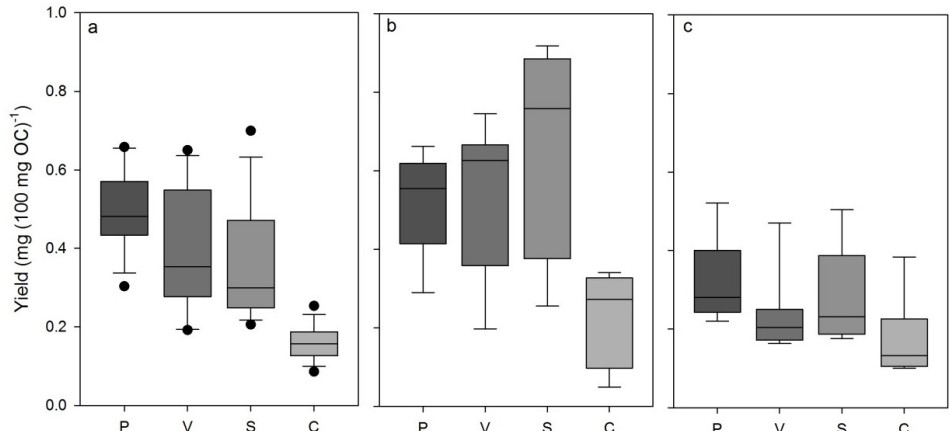


















Figure 7. Pearson correlation between mg DOC lost upon irradiation per mg DOC and a) P:V ratios and b)
Ad:Al$_{v,s}$ (derived from acids and aldehydes from vanillyl and syringyl phenol groups) ratios in all Black Burn
water samples analysed (n=28). Lines of best fit for all water samples are also shown. The monthly samples in
the year-long study and the winter and summer rainfall event samples are indicated.

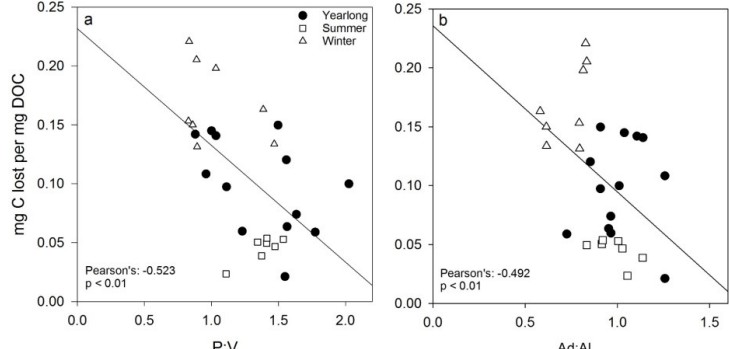


