# Peer review of "Temporal changes in photoreactivity of dissolved organic carbon and implications for aquatic carbon fluxes from peatlands"

_Biogeosciences, 2016_

## Referee Comment (RC1) · P. Neale (Referee) · 9 Sep 2016

Dissolved organic matter (DOM) has been sampled from Scottish peatlands, examining both the upper end stream drainage from a peat bog and from a lake as the lower end receiving basin. This material was experimentally exposed to UV radiation in order to understand DOM photoreactivity and address the hypotheses that photoreactivity is seasonally variable, linked to rainfall events and an important loss term of carbon from the peatland system.

The report is a good contribution to on going efforts within the aquatic biogeochemistry community to better understand the significance of photoreactions to carbon and mineral flows. This is the first study addressing this topic for Scottish peatlands. The sampling and approaches have merit in regard to characterizing seasonality, response to rain events and comparing the systems. Clear patterns of response are resolved for the high DOM site at Black Burn with the interesting result of highest photoreactivity in early winter. On the other hand, DOM in Loch Katrine is much less photoreactive, and a seasonal pattern was not evident although resolution became an issue at the level of responses observed. The authors have made an unusual choice as to irradiation source in the experimental exposures with consequences for the environmental relevance of the production rates and their relation to environmental factors. Unlike any other similar study that I am aware of, the authors chose to a primarily UVB (280-315 nm) emitting fluorescent lamp. This lamp has comparable UVB output as solar irradiance at noontime, on a summer solstice, clear day but much lower UVA (315-400 nm) and PAR (400-700 nm). Thus, most studies on this issue use Xe lamp based solar simulator (example stream study – Porcal et al. 2013) or lamps with primarily UVA output (example Lu et al. 2013). Spectral distribution is important because in most aquatic environments in situ, CDOM will absorb much more UVA than UVB. For relatively "fresh DOM" (using brackish tidal marsh CDOM as an example), about 90% of the absorbed irradiance at the surface is UVA and only 5% UVB (rest PAR). Thus the treatment described could substantially underestimate actual rates exhibited by a sample that experienced an equivalent period of full sun (the experimental 8 h exposure to 1.8 W m-2 is about the same as the cumulative incident UVB on a 14 h cloudless day at solstice, ca. 53 vs 49 kJ m-2 respectively). Potentially, some adjustment for comparability to other studies could be made for this by considering the general shape of the apparent quantum yield spectrum for CO and CO2 photoproduction from DOM comparing the lamp spectrum to solar irradiance (cf. the cited Stubbins et al. and Koehler et al. studies) and in addition by expressing results as a rate constant vs a simple change over the incubation period.

However, there is a larger issue, which is that, as the authors state, due to the effects of bank shading and short transit time of water within the immediate catchment, light driven instream DOC processing is unlikely to be significant for the high DOM Black Burn. Instead, they suggest that the actual processing may occur considerably downstream, in unshaded streams or lentic systems. But the rates there will further depend on the residence time, transparency and optical depth of those systems which are basically unknown for this material. So in the end, I would be very cautious in making any estimate even of an upper bound in the carbon loss rates from these systems given the very substantial methodological bias and involvement of unknown factors. I do agree that given the demonstrated photoreactivity of fresh peatland DOM more work should be done to obtain such an estimate, in particular, if it could somehow be scaled up to a catchment or regional scale.

The choices of irradiation source may also influence the correlation of photoreactivity with other factors, particularly optical characteristics. Several studies have demonstrated that the spectral dependence of absorption and fluorescence photobleaching depends on the spectral distribution of the irradiation source (Del Vecchio and Blough 2002, Tzortziou et al. 2007). UVB-313 fluorescent lamp-based exposure system could produce a distinctly different absorption difference spectrum than natural irradiance (incident or in water), however I do not know of any study that has made the comparison. The results could influence the correlation of photoreactivity and other variables with delta E4:E6, for example. Finally, the spectral distribution of the irradiance source could influence which chromophores are contributing to the mineralization processes for example, which lignin phenols are involved. I do not know whether this is the case, but it is something that should be kept in mind when relating photoreactions to DOM composition.

I made several minor comments on the mss which I have annotated directly on the pdf. On the figures, it would be helpful in visualizing the irradiance induced changes shown in Fig. 4 if independent scales were used for the Black Burn vs Loch Katrine samples. The point that the L. Katrine photoreactivity is much lower won't be lost if (like in the other figures), the difference in scale is called to the attention of the reader. A more important point, is the relative variation in time (or lack thereof) which is presently difficult to see for the L. Katrine results.

Respectfully submitted,

Patrick Neale
Edgewater, MD, USA

[revised manuscript text omitted]

---

## Referee Comment (RC2) · Anonymous Referee #2 · 3 Oct 2016

This study investigates the seasonal and spatial variability in the photoreactivity of DOM from Scottish peatlands. Novel observations are presented on the chemical composition of peatland DOM, the influence of precipitation events on DOM mobilization and the significance of water residence time on DOM photodegradation and export. Peatland systems export high concentrations of photoreactive DOM, and this study demonstrates solar radiation can play an important role in carbon gas fluxes from these systems.

It is important to use lamps that provide a good simulation of sunlight (spectrum and intensity) when investigating photochemical alterations of natural organic matter in the environment. The UV-B 313 lamp used in this study emits short wavelength UV (below

the 295 nm solar cutoff) that is particularly destructive of organic molecules. Therefore, while providing useful information about the relative photoreactivity of DOM among different seasons and locations, the results from this study should not be used to estimate rates of photodegradation in natural waters. In addition, comparisons of the results from this study with those of other studies should be of a qualitative, rather than quantitative, perspective.

A couple of additional optical parameters can provide insights about the source, composition and alteration of DOM. The following parameters should be included: spectral slope (S) 275-295 nm, and the absorption coefficient at 350 nm (a350). The S275-295 is an indicator of DOM molecular weight and extent of photochemical alteration, and the a350 has been used as in indicator of lignin phenol concentrations (Helms et al., 2008; Fichot and Benner 2012).

Specific comments:

Line 48: include Miller and Zepp 1995

Lines 140-141: Filtration (0.22 um) does not exclude microbial activity, it reduces microbial activity (filtered samples contain some active bacteria)

Line 151 – report the wavelength range of light measured by the PMA2102 broad-band sensor

Lines 160-161 – estimation of the exposure time of DOM to solar irradiation needs to consider mixing processes and extinction coefficients for the solar spectrum

Lines 180-182: provide information about the GC column and chromatographic conditions

Line 186: concentrations of DIC were not measured

Line 189: give the pathlength, not volume, of the quartz (?) cuvette

Line 216: provide information about the column and chromatographic conditions

Additional insights about lignin photodegradation can be found in Benner and Kaiser 2011 Biogeochem.. and Lu et al. 2016 Frontiers Mar. Sci.

The clarity of Figures 1, 3, 5, 6 and 7 would be improved by the use of different colors for different parameters

Figure 3: present the change in absorbance as a percentage of the controls and only show the wavelengths starting at 250 nm

Molar units are preferred for all chemical measurements

---

## Author Comment (AC1) · 7 Nov 2016

**Response to comments for the manuscript bg-2016-296 'Temporal changes in photoreactivity of dissolved organic carbon and implications for aquatic carbon fluxes from peatlands'**

Amy E Pickard et al.

November, 2016

We thank the referees for their thorough reading of the manuscript. We address their points (shown in italics) below.

In addition to making the corrections advised by the two referees, we have changed how photoreactivity is measured for seven of the thirteen Loch Katrine samples, where net DOC gains were observed in irradiated sample aliquots. Where DOC gains were observed, photoreactivity (mg C / mg DOC) is now expressed as the sum of gaseous photoproduction (eq.1) divided by the pre-irradiation DOC concentration (eq. 2). For example, in the May 2014 sample:

$$0.07 \text{ mg } CO_2\text{-C L}^{-1} + 0.02 \text{ mg CO-C L}^{-1} = 0.09 \text{ mg C L}^{-1} \qquad \text{(eq. 1)}$$

$$0.09 \text{ mg C L}^{-1} / 5.05 \text{ mg DOC L}^{-1} = 0.02 \text{ mg C mg DOC L}^{-1} \qquad \text{(eq. 2)}$$

This is in contrast to all other water samples, where the sum of gaseous photoproduction is added to net DOC loss (expressed as a positive value) (eq. 3) and divided by the pre-irradiation DOC concentration (eq. 4). For example in the July 2014 sample:

$$0.06 \text{ mg } CO_2\text{-C L}^{-1} + 0.02 \text{ mg CO-C L}^{-1} + 0.46 \text{ mg DOC L}^{-1} = 0.54 \text{ mg C L}^{-1} \quad \text{(eq. 3)}$$

$$0.54 \text{ mg C L}^{-1} / 5.82 \text{ mg DOC L}^{-1} = 0.09 \text{ mg C mg DOC L}^{-1} \qquad \text{(eq. 4)}$$

This method means that no negative photoreactivity values are produced (which may have been explained in large part by the limited resolution of the PPM LabTOC instrument at very low DOC concentrations). Photoreactivity data in figure 4b have been adjusted accordingly.

**Referee 1**

> *Dissolved organic matter (DOM) has been sampled from Scottish peatlands, examining both the upper end stream drainage from a peat bog and from a lake as the lower end receiving basin. This material was experimentally exposed to UV radiation in order to understand DOM photoreactivity and address the hypotheses that photoreactivity is seasonally variable, linked to rainfall events and an important loss term of carbon from the peatland system. The report is a good contribution to on going efforts within the aquatic biogeochemistry community to better understand the significance of photoreactions to carbon and mineral flows. This is the first study addressing this topic for Scottish peatlands. The sampling and approaches have merit in regard to characterizing seasonality, response to rain events and comparing the systems. Clear patterns of response are resolved for the high DOM site at Black Burn with the interesting result of highest photoreactivity in early winter. On the other hand, DOM in Loch Katrine is much less photoreactive, and a seasonal pattern was not evident although resolution became an issue at the level of responses observed.*

We thank reviewer 1 for their positive comments. We believe the edits described below will significantly improve the original manuscript. In particular we have added more discussion about the potential influence of the irradiation source upon the measured results.

> *The authors have made an unusual choice as to irradiation source in the experimental exposures with consequences for the environmental relevance of the production rates and their relation to environmental factors. Unlike any other similar study that I am aware of, the authors chose to a primarily UVB (280-315 nm) emitting fluorescent lamp. This lamp has comparable*

*UVB output as solar irradiance at noontime, on a summer solstice, clear day but much lower UVA (315-400 nm) and PAR (400-700 nm). Thus, most studies on this issue use Xe lamp based solar simulator (example stream study – Porcal et al. 2013) or lamps with primarily UVA output (example Lu et al. 2013). Spectral distribution is important because in most aquatic environments in situ, CDOM will absorb much more UVA than UVB. For relatively "fresh DOM" (using brackish tidal marsh CDOM as an example), about 90% of the absorbed irradiance at the surface is UVA and only 5% UVB (rest PAR). Thus the treatment described could substantially underestimate actual rates exhibited by a sample that experienced an equivalent period of full sun (the experimental 8 h exposure to 1.8 W m-2 is about the same as the cumulative incident UVB on a 14 h cloudless day at solstice, ca. 53 vs 49 kJ m-2 respectively). Potentially, some adjustment for comparability to other studies could be made for this by considering the general shape of the apparent quantum yield spectrum for CO and CO2 photoproduction from DOM comparing the lamp spectrum to solar irradiance (cf. the cited Stubbins et al. and Koehler et al. studies) and in addition by expressing results as a rate constant vs a simple change over the incubation period.*

The irradiation source was selected as UV-B is the most effective source of radiation in producing photochemical effects (Häder et al., 2007; Zepp et al., 2007), however we agree with the reviewer that this choice makes comparison with both previous studies and natural photochemical responses more difficult. In order to allow clearer comparison with previous studies using UV lamp sources, we have now included in the manuscript the following table of weighted action spectra responses for commonly used spectral weighting functions including $CH_4$ from pectin (McLeod et al., 2008), plant growth function (Flint and Caldwell, 2003), general plant action spectrum (Green et al., 1974) and DNA damage (Setlow, 1974). We include this in the method section where details of irradiance regime and output are stated.

**Table 1.** Photosynthetically active radiation (PAR) and ultraviolet irradiances during 8 h exposures to Q-Panel 313 fluorescent lamps filtered with 125 µm cellulose diacetate.

| Irradiance W m$^{-2}$ | | | | | | | |
| --- | --- | --- | --- | --- | --- | --- | --- |
| Total UV (280-400 nm) | UV-A (315-400 nm) | UV-B (280-315 nm) | PAR (400-700 nm) | CH$_4$ [a] | GEN (G) [b] | PG [c] | DNA [d] |
| 7.52 | 4.63 | 2.89 | 0.92 | 2.50 | 1.25 | 1.05 | 0.98 |

[a] $CH_4$, idealized spectral weighting function for $CH_4$ production (McLeod et al. 2008)
[b] weighted with a mathematical function of the general plant action spectrum (Green et al. 1974)
[c] weighted with a the plant growth function of Flint & Caldwell (2003)
[d] weighted with the DNA damage action spectrum (Setlow 1974)

We have also added to the discussion a section on the potential influence of the UV-B 313 lamps on the results of the study and the potential implications the methodology may have for any upscaling attempts.

*However, there is a larger issue, which is that, as the authors state, due to the effects of bank shading and short transit time of water within the immediate catchment, light driven instream DOC processing is unlikely to be significant for the high DOM Black Burn. Instead, they suggest that the actual processing may occur considerably downstream, in unshaded streams or lentic systems. But the rates there will further depend on the residence time, transparency and optical depth of those systems which are basically unknown for this material. So in the end, I would be very cautious in making any estimate even of an upper bound in the carbon loss rates from these systems given the very substantial methodological bias and involvement of unknown factors. I do agree that given the demonstrated photoreactivity of fresh peatland DOM more work should be done to obtain such an estimate, in particular, if it could somehow be scaled up to a catchment or regional scale.*

We agree with the reviewer's comment that there are significant uncertainties in downstream DOC turnover and have adopted more cautious wording in our discussion section 'Implications for photochemical turnover of DOC in aquatic systems'. We have also removed the potential evaded photochemical $CO_2$ estimate and instead suggest that given the significant volume of DOC produced by the catchment, in-stream photo-processing may be an important term in carbon budgets of peatland draining aquatic systems.

> *The choices of irradiation source may also influence the correlation of photoreactivity with other factors, particularly optical characteristics. Several studies have demonstrated that the spectral dependence of absorption and fluorescence photobleaching depends on the spectral distribution of the irradiation source (Del Vecchio and Blough 2002, Tzortziou et al. 2007). UVB-313 fluorescent lamp-based exposure system could produce a distinctly different absorption difference spectrum than natural irradiance (incident or in water), however I do not know of any study that has made the comparison. The results could influence the correlation of photoreactivity and other variables with delta E4:E6, for example.*

We thank the reviewer for highlighting this issue. We have added text to discussion section 'Implications for photochemical turnover of DOC in aquatic systems' which evaluates the possible influence that the irradiation source (UV-B 313 lamps) may have had on the optical characteristics of water samples. Correlations between delta E4:E6 values and other variables in Table 2 (now Table 3) remain in the manuscript as we assert that because all samples were exposed to the same irradiation conditions, the relative differences in the values can provide interesting information pertaining to factors influencing carbon budget changes.

> *Finally, the spectral distribution of the irradiance source could influence which chromophores are contributing to the mineralization processes for example, which lignin phenols are involved. I do not know whether this is the case, but it is something that should be kept in mind when relating photoreactions to DOM composition.*

We agree that this would be an interesting line of enquiry. However, given the lack of literature on the topic we feel that it would be difficult to discuss the potential influence of spectral distribution of the irradiation source on preferential degradation of phenol groups. In our study lignin phenols were not measured in irradiated samples and hence we could not support such discussion with any evidence.

> *I made several minor comments on the mss which I have annotated directly on the pdf. On the figures, it would be helpful in visualizing the irradiance induced changes shown in Fig. 4 if independent scales were used for the Black Burn vs Loch Katrine samples. The point that the L. Katrine photoreactivity is much lower won't be lost if (like in the other figures), the difference in scale is called to the attention of the reader. A more important point, is the relative variation in time (or lack thereof) which is presently difficult to see for the L. Katrine results.*

All the minor comments annotated directly on the pdf have been addressed. We have adjusted the scale of the plots for the Loch Katrine results, and noted the difference in scale in the figure caption. We have also adjusted the method for determining photoreactivity in Loch Katrine samples where net DOC gains were observed upon irradiation, as explained on p.1 of the author comment above.

**Referee 2**

> *This study investigates the seasonal and spatial variability in the photoreactivity of DOM from Scottish peatlands. Novel observations are presented on the chemical composition of peatland DOM, the influence of precipitation events on DOM mobilization and the significance of water residence time on DOM photodegradation and export. Peatland systems export high concentrations of photoreactive DOM, and this study demonstrates solar radiation can play an important role in carbon gas fluxes from these systems.*

We thank the reviewer for their positive comments and for their constructive criticism of the manuscript.

*It is important to use lamps that provide a good simulation of sunlight (spectrum and intensity) when investigating photochemical alterations of natural organic matter in the environment. The UV-B 313 lamp used in this study emits short wavelength UV (below the 295 nm solar cutoff) that is particularly destructive of organic molecules. Therefore, while providing useful information about the relative photoreactivity of DOM among different seasons and locations, the results from this study should not be used to estimate rates of photodegradation in natural waters. In addition, comparisons of the results from this study with those of other studies should be of a qualitative, rather than quantitative, perspective.*

The UV-B 313 lamps were covered with a film of 125 μm cellulose diacetate (CD), as described in the method section of the original manuscript. However we made a typographical error when stating the transmission properties of the CD film. CD provides a cut-off point at 290 nm, below which no light is transmitted (e.g. McLeod et al., 2008; Fraser et al., 2015). This has now been corrected in the manuscript and should assure the reviewer that there are very limited photochemical effects generated as a function of short wavelength UV-B which is not present in the natural solar spectrum.

In alignment with this comment and similar comments provided by the first reviewer, we have removed the potential evaded photochemical $CO_2$ estimate and instead suggest that given the significant volume of DOC produced by the catchment, in-stream processing may be an important term in carbon budgets of peatland draining aquatic systems.

Comparisons of percentage DOC losses to other photodegradation studies cited in the discussion section 'Peatlands as a source of photochemically labile DOC' have been retained in the text. We believe that the inclusion in the manuscript of figures from other studies will give the reader confidence that although a UV-B irradiation source was used in this study, the magnitude of photochemically induced DOC losses are comparable to previous studies which used a solar simulator to output a natural irradiation spectrum.

*A couple of additional optical parameters can provide insights about the source, composition and alteration of DOM. The following parameters should be included: spectral slope (S) 275-295 nm, and the absorption coefficient at 350 nm (a350). The S275-295 is an indicator of DOM molecular weight and extent of photochemical alteration, and the a350 has been used as in indicator of lignin phenol concentrations (Helms et al., 2008; Fichot and Benner 2012).*

Thanks for these suggestions. We have incorporated both parameters into Table 2 of the manuscript and have included a method description for the spectral slope calculation.

*Specific comments:*

*Line 48: include Miller and Zepp 1995*

Included.

*Lines 140-141: Filtration (0.22 um) does not exclude microbial activity, it reduces microbial activity (filtered samples contain some active bacteria)*

Sentence corrected to: "…syringe driven pore size filters 0.22 μm (Merck Millipore, UK) to reduce the effect of microbial activity".

*Line 151 – report the wavelength range of light measured by the PMA2102 broad-band sensor*

We have stated that the wavelength range is within the UV-B and that the sensor is erythemally weighted which allows comparison with previous studies through use of spectral weighting functions.

*Lines 160-161 – estimation of the exposure time of DOM to solar irradiation needs to consider mixing processes and extinction coefficients for the solar spectrum*

We believe that this information is more relevant in the discussion and have now explicitly alluded to difficulties in estimating DOM exposure due to mixing processes in the section 'Implications for photochemical turnover of DOC in aquatic systems'.

*Lines 180-182: provide information about the GC column and chromatographic conditions*

We have added further information pertaining to GC analysis, including sample size, needle penetration depth and analytical run length.

*Line 186: concentrations of DIC were not measured*

Corrected to "DOC and TC were measured…"

*Line 189: give the pathlength, not volume, of the quartz (?) cuvette*

Pathlength information provided. We used disposable PLASTIBRAND® UV-Cuvettes for our analyses. This information has also been added to the method.

*Line 216: provide information about the column and chromatographic conditions*

We have added further information pertaining to the GC column and chromatographic conditions.

*Additional insights about lignin photodegradation can be found in Benner and Kaiser 2011 Biogeochem.. and Lu et al. 2016 Frontiers Mar. Sci.*

Thanks for these paper recommendations. We have referenced them in the text.

*The clarity of Figures 1, 3, 5, 6 and 7 would be improved by the use of different colors for different parameters*

We have changed the figures and improved clarity by adopting a consistent colour palette.

*Figure 3: present the change in absorbance as a percentage of the controls and only show the wavelengths starting at 250 nm*

Adjusted as specified.

*Molar units are preferred for all chemical measurements*

We have retained the original units with concentrations expressed in mg L$^{-1}$ or μg L$^{-1}$, as this format allows us to directly compare results with both previous studies at the Auchencorth Moss catchment (Dinsmore et al., 2010, 2013) and other relevant research in the field of DOC processing (e.g. Moody et al., 2013; Palmer et al., 2015; Spencer et al., 2009).

**References**

Dinsmore, K. J., Billett, M. F., Skiba, U. M., Rees, R. M., Drewer, J. and Helfter, C.: Role of the aquatic pathway in the carbon and greenhouse gas budgets of a peatland catchment, Glob. Chang. Biol., 16, 2750–2762, doi:10.1111/j.1365-2486.2009.02119.x, 2010.

Dinsmore, K. J., Billett, M. F. and Dyson, K. E.: Temperature and precipitation drive temporal variability in aquatic carbon and GHG concentrations and fluxes in a peatland catchment, Glob. Chang. Biol., 19, 2133–2148, doi:10.1111/gcb.12209, 2013.

Flint, S. D. and Caldwell, M. M.: A biological spectral weighting function for ozone depletion research with higher plants, Physiol. Plant., 117, 137–144, doi:10.1034/j.1399-3054.2003.1170117.x, 2003.

Fraser, W. T., Blei, E., Fry, S. C., Newman, M. F., Reay, D. S., Smith, K. A. and McLeod, A. R.: Emission of methane, carbon monoxide, carbon dioxide and short-chain hydrocarbons from vegetation foliage under ultraviolet irradiation, Plant, Cell Environ., 38(5), 980–989, doi:10.1111/pce.12489, 2015.

Green, A. E. S., Sawada, T. and Shettle, E. P.: The middle ultraviolet reaching the ground, Photochem. Photobiol., 19(4), 251–259, doi:10.1111/j.1751-1097.1974.tb06508.x, 1974.

Häder, D.-P., Kumar, H. D., Smith, R. C. and Worrest, R. C.: Effects of solar UV radiation on aquatic ecosystems and interactions with climate change., Photochem. Photobiol. Sci., 6(3), 267–85, doi:10.1039/b700020k, 2007.

McLeod, A. R., Fry, S. C., Loake, G. J., Messenger, D. J., Reay, D. S., Smith, K. A. and Yun, B.-W.: Ultraviolet radiation drives methane emissions from terrestrial plant pectins., New Phytol., 180, 124–132, doi:10.1111/j.1469-8137.2008.02571.x, 2008.

Moody, C. S., Worrall, F., Evans, C. D. and Jones, T. G.: The rate of loss of dissolved organic carbon (DOC) through a catchment, J. Hydrol., 492, 139–150, doi:10.1016/j.jhydrol.2013.03.016, 2013.

Palmer, S. M., Evans, C. D., Chapman, P. J., Burden, A., Jones, T. G., Allott, T. E. H., Evans, M. G., Moody, C. S., Worrall, F. and Holden, J.: Sporadic hotspots for physico-chemical retention of aquatic organic carbon: from peatland headwater source to sea, Aquat. Sci., 1–14, doi:10.1007/s00027-015-0448-x, 2015.

Setlow, R. B.: The wavelengths in sunlight effective in producing skin cancer: a theoretical analysis., Proc. Natl. Acad. Sci. U. S. A., 71(9), 3363–3366, doi:10.1073/pnas.71.9.3363, 1974.

Spencer, R.G.M., Stubbins, A., Hernes, P.J., Baker, A., Mopper, K., Aufdenkampe, A.K., Dyda, R.Y., Mwamba, Vi.L. Mangangu, A.M., Wabakanghanzi, J.N., Six, J.: Photochemical degradation of dissolved organic matter and dissolved lignin phenols from the Congo River, J. Geophys. Res. Biogeosciences, 114(3), 1–12, doi:10.1029/2009JG000968, 2009.

Worrall, F., Burt, T.P., Adamson, J.: The rate of and controls upon DOC loss in a peat catchment, J. Hydrol., 321(1–4), 311–325, 2006.

Zepp, R. G., Erickson, D. J., Paul, N. D. and Sulzberger, B.: Interactive effects of solar UV radiation and climate change on biogeochemical cycling., Photochem. Photobiol. Sci., 6, 286–300, doi:10.1039/b700021a, 2007.

---

## Author Response (AR1)

**Response to comments for the manuscript bg-2016-296 'Temporal changes in photoreactivity of dissolved organic carbon and implications for aquatic carbon fluxes from peatlands'**

Amy E Pickard et al.

February, 2017

We thank the referees for their thorough reading of the manuscript. We address their points and those of the Associate Editor (shown in italics) below.

In addition to making the corrections advised by the two referees and associate editor, we have changed how photoreactivity is quantified in the Loch Katrine samples, where minimal net DOC changes upon irradiation were observed in the sample aliquots. Photoreactivity (mg C / mg DOC) is now expressed as the sum of gaseous photoproduction divided by the pre-irradiation DOC concentration (eq. 1). For example, in the Loch Katrine May 2014 sample:

$$\text{Photoreactivity} = \frac{\left(0.07 \text{ mg CO}_2\text{-C L}^{-1} + 0.02 \text{ mg CO-C L}^{-1}\right)}{5.05 \text{ mg DOC L}^{-1}} = 0.02 \text{ mg C /mg DOC} \quad \text{(eq. 1)}$$

This is in contrast to Black Burn water samples, where the sum of gaseous photoproduction is added to net DOC loss (expressed as a positive value) and divided by the pre-irradiation DOC concentration (eq. 2). For example in the Black Burn May 2014 sample:

$$\text{Photoreactivity} = \frac{\left(1.62 \text{ mg CO}_2\text{-C L}^{-1} + 0.13 \text{ mg CO-C L}^{-1} + 2.02 \text{ mg DOC L}^{-1}\right)}{50.9 \text{ mg DOC L}^{-1}}$$

$$= 0.07 \text{ mg C /mg DOC} \quad \text{(eq. 2)}$$

This method means that no negative photoreactivity values are produced for Loch Katrine (which may have been explained in large part by the limited resolution of the PPM LabTOC instrument at very low DOC concentrations). Photoreactivity data in figure 4b have been adjusted accordingly and text explaining the revised photoreactivity calculations has been added to the data analysis section 2.6 (lines 256-262). The revised method for calculating the photoreactivity of the Loch Katrine samples meant that the mean spring absorbance values shown in Figure 3b are now very similar to those for winter and autumn. Consequently the text referring to the previously apparent difference has been removed from Discussion section 4.1.

**Associate Editor comments**

> *Your manuscript has been reviewed by two referees. Both referees agreed that the manuscript represents a substantial contribution to scientific progress within the scope of this journal, and that the measurements discussed are novel and provide new insights on the chemical composition and photochemical reactivity of peatland DOM. At the same time, the reviewers recommended a number of improvements. Both referees were concerned about the choice of irradiation source, how relevant this study's findings are to photochemical transformation processes in natural waters and how comparable the results are, in quantitative terms, with those of other studies. The implications of the choice of irradiation source would need to be discussed in detail in the revised manuscript, also in the context of previous studies discussing the dependence of absorption and fluorescence photobleaching on the spectral quality of exposure.*

We thank the editor for their positive comments and for their suggestions to improve the manuscript. In alignment with these comments, we have added significant text to the discussion to highlight that the UV exposure used here allows only for relative differences over time and between our sites to be elucidated (lines 412-420). Furthermore we suggest that the spectral dependence of absorption may mean that changes in for example the E4:E6 ratio may not be comparable to other studies and that its use in correlation analyses should be interpreted with caution (lines 439-448).

In addition to the discussion changes we have also added a spectral weighting function table to the methods section to compare the weighted exposure used in this study to standard weighting functions (Table 1).

> *In addition, the reviewers recommended to show additional parameters to get more insights on the composition and transformation of DOM in the study region. Additional information, from ancillary measurements or existing literature, on downstream water composition and light attenuation/optical depth could also help address the question raised by the reviewers regarding downstream DOC processing/turnover rates.*

We have added additional parameters that provide insights to DOM transformation to our paper, including changes to absorbance at 350 nm and the spectral slope of absorbance. These have been included in the correlation analyses shown in Table 3. In terms of additional parameters to estimate downstream DOC processing rates, we believe that without significant further data collection obtaining a reasonable estimate is beyond the scope of this study. Instead we have expanded the discussion of the uncertainties in downstream DOC processing and have suggested that future studies should focus attention on potential hotspots for DOC processing (lines 529-558), which include mixing zones of freshwaters with different pH, conductivity and metal concentrations as identified by Palmer et al. (2015) in their study of peatland headwaters.

> *As mentioned by Reviewer #2, filtration through 0.22 um does not completely remove bacteria from the sample. The authors should report in their revised manuscript whether during their experiments they observed any changes (with time) in CDOM or DOC in their "control" samples (0.22 um, not exposed to light). On line 286, the manuscript mentions that: "dark control samples showed a greater drop in absorbance upon irradiation than light exposed samples". More discussion is needed on these results.*

We thank the editor for highlighting this point. We have amended our text to make clear that the absorbance increases occurred in the irradiated samples relative to the initial absorbance values measured prior to the experiment (lines 305-306). Control samples showed no change relative to the initial samples. We have added a section to the discussion which reviews the potential effect of bacterial DOC production in the UV exposed samples in summer and suggest that carbon isotope data may help to resolve such uncertainties (lines 475-482).

**Referee 1**

> *Dissolved organic matter (DOM) has been sampled from Scottish peatlands, examining both the upper end stream drainage from a peat bog and from a lake as the lower end receiving basin. This material was experimentally exposed to UV radiation in order to understand DOM photoreactivity and address the hypotheses that photoreactivity is seasonally variable, linked to rainfall events and an important loss term of carbon from the peatland system. The report is a good contribution to on going efforts within the aquatic biogeochemistry community to better understand the significance of photoreactions to carbon and mineral flows. This is the first study addressing this topic for Scottish peatlands. The sampling and approaches have merit in regard to characterizing seasonality, response to rain events and comparing the systems. Clear patterns of response are resolved for the high DOM site at Black Burn with the interesting result of highest photoreactivity in early winter. On the other hand, DOM in Loch Katrine is much less photoreactive, and a seasonal pattern was not evident although resolution became an issue at the level of responses observed.*

We thank reviewer 1 for their positive comments. We believe the edits described below will significantly improve the original manuscript. In particular we have added more discussion about the potential influence of the irradiation source upon the measured results.

> *The authors have made an unusual choice as to irradiation source in the experimental exposures with consequences for the environmental relevance of the production rates and their relation to environmental factors. Unlike any other similar study that I am aware of, the authors chose to a primarily UVB (280-315 nm) emitting fluorescent lamp. This lamp has comparable UVB output as solar irradiance at noontime, on a summer solstice, clear day but much lower UVA (315-400 nm) and PAR (400-700 nm). Thus, most studies on this issue use Xe lamp based solar simulator (example stream study – Porcal et al. 2013) or lamps with primarily UVA output (example Lu et al. 2013). Spectral distribution is important because in most aquatic environments in situ, CDOM will absorb much more UVA than UVB. For relatively "fresh DOM" (using brackish tidal marsh CDOM as an example), about 90% of the absorbed irradiance at the surface is UVA and only 5% UVB (rest PAR). Thus the treatment described could substantially underestimate actual rates exhibited by a sample that experienced an equivalent period of full sun (the experimental 8 h exposure to 1.8 W m-2 is about the same as the cumulative incident UVB on a 14 h cloudless day at solstice, ca. 53 vs 49 kJ m-2 respectively). Potentially, some adjustment for comparability to other studies could be made for this by considering the general shape of the apparent quantum yield spectrum for CO and CO2 photoproduction from DOM comparing the lamp spectrum to solar irradiance (cf. the cited Stubbins et al. and Koehler et al. studies) and in addition by expressing results as a rate constant vs a simple change over the incubation period.*

The irradiation source was selected as UVB is the most effective source of radiation in producing photochemical effects (Häder et al., 2007; Zepp et al., 2007), however we agree with the reviewer that this choice makes comparison with both previous studies and natural photochemical responses more difficult. In order to allow clearer comparison with previous studies using UV lamp sources, we have now included in the manuscript the following table of weighted action spectra responses for commonly used spectral weighting functions including $CH_4$ from pectin (McLeod et al., 2008), plant growth function (Flint and Caldwell, 2003), general plant action spectrum (Green et al., 1974) and DNA damage (Setlow, 1974). We include this in the methods section where details of irradiance regime and output are stated (lines 154-160).

**Table 1.** Photosynthetically active radiation (PAR) and ultraviolet irradiances during 8 h exposures to Q-Panel 313 fluorescent lamps filtered with 125 µm cellulose diacetate.

| Irradiance W m$^{-2}$ | | | | | | | |
|---|---|---|---|---|---|---|---|
| Total UV (280-400 nm) | UV-A (315-400 nm) | UV-B (280-315 nm) | PAR (400-700 nm) | $CH_4$ [a] | GEN (G) [b] | PG [c] | DNA [d] |
| 7.52 | 4.63 | 2.89 | 0.92 | 2.50 | 1.25 | 1.05 | 0.98 |

[a] $CH_4$, idealized spectral weighting function for $CH_4$ production (McLeod et al. 2008)
[b] weighted with a mathematical function of the general plant action spectrum (Green et al. 1974)
[c] weighted with the plant growth (PG) function (Flint & Caldwell 2003)
[d] weighted with the DNA damage action spectrum (Setlow 1974)

We have also added to the discussion a section on the potential influence of the UVB313 lamps on the results of the study and the potential implications the methodology may have for any upscaling attempts (lines 412-420).

> *However, there is a larger issue, which is that, as the authors state, due to the effects of bank shading and short transit time of water within the immediate catchment, light driven instream*

*DOC processing is unlikely to be significant for the high DOM Black Burn. Instead, they suggest that the actual processing may occur considerably downstream, in unshaded streams or lentic systems. But the rates there will further depend on the residence time, transparency and optical depth of those systems which are basically unknown for this material. So in the end, I would be very cautious in making any estimate even of an upper bound in the carbon loss rates from these systems given the very substantial methodological bias and involvement of unknown factors. I do agree that given the demonstrated photoreactivity of fresh peatland DOM more work should be done to obtain such an estimate, in particular, if it could somehow be scaled up to a catchment or regional scale.*

We agree with the reviewer's comment that there are significant uncertainties in downstream DOC turnover and have adopted more cautious wording in our discussion section 'Implications for photochemical turnover of DOC in aquatic systems'. We have also removed the potential evaded photochemical $CO_2$ estimate and instead suggest that given the significant volume of DOC produced by the catchment, in-stream photo-processing may be an important term in carbon budgets of peatland draining aquatic systems (lines 529-539).

*The choices of irradiation source may also influence the correlation of photoreactivity with other factors, particularly optical characteristics. Several studies have demonstrated that the spectral dependence of absorption and fluorescence photobleaching depends on the spectral distribution of the irradiation source (Del Vecchio and Blough 2002, Tzortziou et al. 2007). UVB-313 fluorescent lamp-based exposure system could produce a distinctly different absorption difference spectrum than natural irradiance (incident or in water), however I do not know of any study that has made the comparison. The results could influence the correlation of photoreactivity and other variables with delta E4:E6, for example.*

We thank the reviewer for highlighting this issue. We have added text to Discussion section 4.1 'Peatlands as a source of photochemically labile DOC' which evaluates the possible influence that the irradiation source (UV313 lamps) may have had on the optical characteristics of water samples and suggests that results should be interpreted with caution (lines 439-448). Correlations between delta E4:E6 values and other variables in Table 2 (now Table 3) remain in the manuscript as we assert that because all samples were exposed to the same irradiation conditions, the relative differences in the values can provide interesting information pertaining to factors influencing carbon budget changes.

*Finally, the spectral distribution of the irradiance source could influence which chromophores are contributing to the mineralization processes for example, which lignin phenols are involved. I do not know whether this is the case, but it is something that should be kept in mind when relating photoreactions to DOM composition.*

We agree that this would be an interesting line of enquiry. However, given the lack of literature on the topic we feel that it would be difficult to discuss the potential influence of spectral distribution of the irradiation source on preferential degradation of phenol groups. In our study lignin phenol analysis was not conducted on irradiated samples and hence we could not support such discussion with any evidence.

*I made several minor comments on the mss which I have annotated directly on the pdf. On the figures, it would be helpful in visualizing the irradiance induced changes shown in Fig. 4 if independent scales were used for the Black Burn vs Loch Katrine samples. The point that the L. Katrine photoreactivity is much lower won't be lost if (like in the other figures), the difference in scale is called to the attention of the reader. A more important point, is the relative variation in time (or lack thereof) which is presently difficult to see for the L. Katrine results.*

All the minor comments annotated directly on the pdf have been addressed. We have adjusted the y-axis scale in Fig. 4 for plots for the Loch Katrine samples, and noted the difference in scale in the figure caption. We have also adjusted the method for determining photoreactivity in Loch Katrine samples where net DOC gains were observed upon irradiation, as explained on pp.1 of the author comment above.

**Referee 2**

> *This study investigates the seasonal and spatial variability in the photoreactivity of DOM from Scottish peatlands. Novel observations are presented on the chemical composition of peatland DOM, the influence of precipitation events on DOM mobilization and the significance of water residence time on DOM photodegradation and export. Peatland systems export high concentrations of photoreactive DOM, and this study demonstrates solar radiation can play an important role in carbon gas fluxes from these systems.*

We thank the reviewer for their positive comments and for their constructive criticism of the manuscript.

> *It is important to use lamps that provide a good simulation of sunlight (spectrum and intensity) when investigating photochemical alterations of natural organic matter in the environment. The UV-B 313 lamp used in this study emits short wavelength UV (below the 295 nm solar cutoff) that is particularly destructive of organic molecules. Therefore, while providing useful information about the relative photoreactivity of DOM among different seasons and locations, the results from this study should not be used to estimate rates of photodegradation in natural waters. In addition, comparisons of the results from this study with those of other studies should be of a qualitative, rather than quantitative, perspective.*

The UV313 lamps were covered with a film of 125 µm cellulose diacetate (CD), as described in the methods section of the original manuscript. However we made a typographical error when stating the transmission properties of the CD film. CD provides a cut-off point at 290 nm, below which no light is transmitted (e.g. McLeod et al., 2008; Fraser et al., 2015). This has now been corrected in the manuscript (lines 144-146 and 416-420) and should assure the reviewer that there are very limited photochemical effects generated as a function of short wavelength UV which is not present in the natural solar spectrum.

In alignment with this comment and similar comments provided by the first reviewer, we have removed the potential evaded photochemical $CO_2$ estimate and instead suggest that, given the significant volume of DOC produced by the catchment, in-stream processing may be an important term in carbon budgets of peatland draining aquatic systems (lines 529-539).

Comparisons of percentage DOC losses to other photodegradation studies cited in the discussion section 'Peatlands as a source of photochemically labile DOC' have been retained in the text. We believe that the inclusion in the manuscript of figures from other studies will give the reader confidence that although a UV-B irradiation source was used in this study, the magnitude of photochemically induced DOC losses are comparable to previous studies which used a solar simulator to output a natural irradiation spectrum.

> *A couple of additional optical parameters can provide insights about the source, composition and alteration of DOM. The following parameters should be included: spectral slope (S) 275-295 nm, and the absorption coefficient at 350 nm (a350). The S275-295 is an indicator of DOM molecular weight and extent of photochemical alteration, and the a350 has been used as in indicator of lignin phenol concentrations (Helms et al., 2008; Fichot and Benner 2012).*

Thanks for these suggestions. We have incorporated both parameters into Table 2 of the manuscript and have included a method description for the spectral slope calculation (lines 204-207).

*Specific comments:*

> *Line 48: include Miller and Zepp 1995*

Included (see line 46 and 718-720).

*Lines 140-141: Filtration (0.22 um) does not exclude microbial activity, it reduces microbial activity (filtered samples contain some active bacteria)*

Sentence corrected to: "…syringe driven pore size MCE filters 0.22 μm (Merck Millipore, UK) to reduce the effect of microbial activity" (lines 137-139).

*Line 151 – report the wavelength range of light measured by the PMA2102 broad-band sensor*

We have stated that the wavelength range covers both UVA and UVB and that the sensor is erythemally weighted which allows comparison with previous studies through use of spectral weighting functions (lines 148-160).

*Lines 160-161 – estimation of the exposure time of DOM to solar irradiation needs to consider mixing processes and extinction coefficients for the solar spectrum*

We believe that this information is more relevant in the discussion and have now explicitly alluded to difficulties in estimating DOM exposure due to mixing processes in the Discussion section 4.3 'Implications for photochemical turnover of DOC in aquatic systems' (lines 529-539 and 552-558).

*Lines 180-182: provide information about the GC column and chromatographic conditions*

We have added further information pertaining to GC analysis, including sample size, needle penetration depth and analytical run length (lines 179-190).

*Line 186: concentrations of DIC were not measured*

Corrected to "DOC and TC were measured…" (line 191)

*Line 189: give the pathlength, not volume, of the quartz (?) cuvette*

Pathlength information provided. We used disposable PLASTIBRAND® UV-Cuvettes for our analyses. This information has also been added to the method (line 194).

*Line 216: provide information about the column and chromatographic conditions*

We have added further information pertaining to the GC column and chromatographic conditions (lines 225-231).

*Additional insights about lignin photodegradation can be found in Benner and Kaiser 2011 Biogeochem. and Lu et al. 2016 Frontiers Mar. Sci.*

Thanks for these paper recommendations. We have referenced them in the text (lines 492, 504, 522, 595-597, 698-700).

*The clarity of Figures 1, 3, 5, 6 and 7 would be improved by the use of different colors for different parameters*

We have changed the figures and improved clarity by adopting a consistent colour palette.

*Figure 3: present the change in absorbance as a percentage of the controls and only show the wavelengths starting at 250 nm*

Adjusted as specified. We have chosen to show data from 250 – 400 nm in the revised figure, as percentage absorbance data became noisy for the Loch Katrine samples in the visible part of the spectrum (>400 nm).

*Molar units are preferred for all chemical measurements*

After consideration we have decided to retain the original units with concentrations expressed in $mg\,L^{-1}$ or $\mu g\,L^{-1}$, as this format allows comparison of results with both previous studies at the Auchencorth Moss catchment (Dinsmore et al., 2010, 2013) and other relevant research in the field of DOC processing (e.g. Moody et al., 2013; Palmer et al., 2015; Spencer et al., 2009).

Lu, C. hia-Jung., Benner, R.onald, Fichot, C.édric G., Fukuda, H.ideki, Yamashita, Y.ouhei, and

Ogawa, O. H.: Sources and tTransformations of dDissolved Llignin Pphenols and Cchromophoric Ddissolved Ooorganic Mmatter 
[revised manuscript text omitted]